# ON SCALING UP 3D GAUSSIAN SPLATTING TRAINING

**Hexu Zhao**[1], **Haoyang Weng**[1,*] **Daohan Lu**[1,*] **Ang Li**[2], **Jinyang Li**[1], **Aurojit Panda**[1], **Saining Xie**[1]
[1]New York University
[2]Pacific Northwest National Laboratory

## ABSTRACT

3D Gaussian Splatting (3DGS) is increasingly popular for 3D reconstruction due to its superior visual quality and rendering speed. However, 3DGS training currently occurs on a single GPU, limiting its ability to handle high-resolution and large-scale 3D reconstruction tasks due to memory constraints. We introduce Grendel, a distributed system designed to partition 3DGS parameters and parallelize computation across multiple GPUs. As each Gaussian affects a small, dynamic subset of rendered pixels, Grendel employs sparse all-to-all communication to transfer the necessary Gaussians to pixel partitions and performs dynamic load balancing. Unlike existing 3DGS systems that train using one camera view image at a time, Grendel supports batched training with multiple views. We explore various optimization hyperparameter scaling strategies and find that a simple *sqrt(batch_size)* scaling rule is highly effective. Evaluations using large-scale, high-resolution scenes show that Grendel enhances rendering quality by scaling up 3DGS parameters across multiple GPUs. On the 4K "Rubble" dataset, we achieve a test PSNR of 27.28 by distributing 40.4 million Gaussians across 16 GPUs, compared to a PSNR of 26.28 using 11.2 million Gaussians on a single GPU. Grendel is an open-source project available at: `https://github.com/nyu-systems/Grendel-GS`

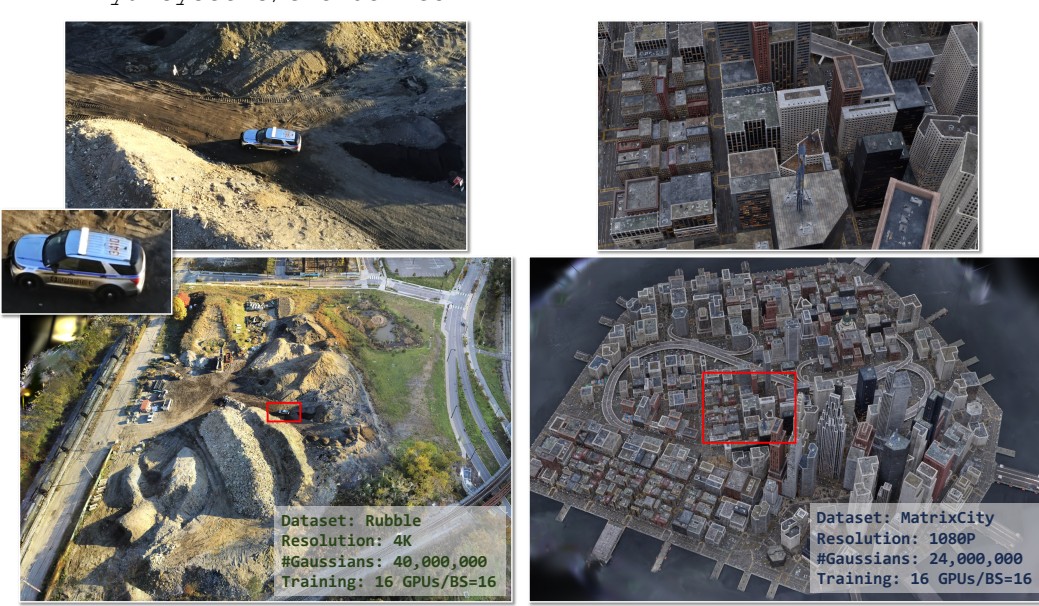

Figure 1: Two large-scale, high-resolution scene reconstructions using Grendel, our distributed 3D Gaussian rendering system. Both images are rendered using 16 GPUs. The left and right images are represented using 40 million and 24 million gaussians respectively. Grendel achieves state of the art quality (PSNR) for both scenes.

## 1 INTRODUCTION

3D Gaussian Splatting (Kerbl et al., 2023) (3DGS) has emerged as a popular technique for 3D novel view synthesis, primarily due to its faster training and rendering compared to previous approaches such as NeRF (Mildenhall et al., 2020). However, most existing 3DGS pipelines are constrained to using a single GPU for training, creating memory and computation bottlenecks when applied

---

[*]Daohan and Haoyang contributed equally to this work.

to high-resolution or larger-scale scenes. For example, the standard Rubble dataset (Turki et al., 2022) contains 1657 images, each with a 4K resolution. A single A100 40GB GPU can hold up to 11.2 million Gaussians – well below the quality saturation point for 3DGS. As we demonstrate in Section 5.2.1, increasing the number of Gaussians continues to improve reconstruction quality. Therefore, in order to *scale up* 3DGS training in terms of parameter count and speed, we develop the Grendel system, which distributes 3DGS training across multiple GPUs and uses an empirical rule to automatically adapt training hyperparameters based on the batch size.

Existing work on scaling 3DGS to large-scale scenes have explored approaches such as divide-and-conquer Lin et al. (2024); Liu et al. (2024); Kerbl et al. (2024), level-of-detail representation and rendering (Kerbl et al., 2024; Ren et al., 2024; Lu et al., 2024). These methods are complementary to the system-level parallelization approach that we adopt to scale 3DGS beyond the memory and compute limitations of a single GPU.

As distributed training frameworks have become widely used for many state-of-the-art DNN models such as LLMs (Shoeybi et al., 2020; Rajbhandari et al., 2020), it is tempting to also use them to distribute 3DGS. However, although 3DGS uses gradient-based optimization, it is *not* based on neural networks. Specifically, it features a unique computation pipeline with dynamic and imbalanced workload patterns. Consequently, existing DNN training frameworks (Shoeybi et al., 2020; Rajbhandari et al., 2020; Li et al., 2020; Zhao et al., 2023), which assume consistent and balanced workload with regular dense tensor operations, would not work well for 3DGS.

In this paper, we present several key observations on scaling up 3DGS that inform the design of our distributed training pipeline. For instance, we note that each stage of the 3DGS training pipeline in an iteration can be effectively parallelized, but the axes of parallelization differ across stages, resulting in *mixed parallelism*. More concretely, in 3DGS, some computations operate on individual output pixels (allowing for pixel-wise parallelism), while others operate on individual 3D Gaussians (allowing for Gaussian-wise parallelism). Mixed parallelism necessitates data shuffling between stages. To minimize communication, we also observe that 3DGS exhibits *spatial locality*, where only a small number of Gaussians affect the rendering of each output image patch. Finally, the computational intensity of rendering an output pixel changes as training progresses. Such *dynamic and unbalanced workloads* will cause any static workload partitioning strategy to become suboptimal.

In this paper, we describe Grendel, a distributed 3DGS training framework designed to leverage our above observations. Grendel uses Gaussian-wise distribution–that is, it distributes Gaussians across GPUs–for steps in a training iteration that exhibit Gaussian-wise parallelism, and pixel-wise distribution for other steps. It minimizes the communication overhead when switching between Gaussian-wise and pixel-wise distribution by assigning contiguous image areas to GPUs during pixel-wise distribution and exploiting spatial locality to minimize the number of Gaussians transferred among GPUs. Finally, Grendel employs a dynamic load balancer that uses previous training iterations to distribute pixel-wise computations to minimize workload imbalance.

Grendel additionally scales up training by batching multiple images. This differs from conventional 3DGS training that exclusively uses a batch size of 1, which would lead to reduced GPU utilization in our distributed framework. To maintain data efficiency and reconstruction quality with larger batches, one needs to re-tune optimizer hyperparameters. To this end, we introduce an automatic hyperparameter scaling rule for batched 3DGS training based on a heuristical *independent gradients hypothesis*. We empirically validate the effectiveness of our proposed approach — Grendel supports distributed training with large batch sizes (we test up to 32) while maintaining reconstruction quality and data efficiency compared to batch size $= 1$.

In summary, our work makes the following contributions:

- We describe the design and implementation of Grendel, a scalable, memory-efficient, adaptive distributed training system for 3DGS. Grendel allows batched 3DGS training to be scaled up and run on up to 32 GPUs.
- We explore the large-batch training dynamics of 3DGS to identify a simple *sqrt(batch_size)* learning rate scaling strategy, which enables efficient, hyperparameter-tuning-free training for batch sizes beyond one.
- We show that Grendel enables high-resolution large scale scene rendering: we use 16 GPUs and render 4K images for large-scale Rubble scene from MegaNERF (Turki et al., 2022). For this scene, Grendel uses 40.4 million Gaussians to achieve a PSNR of 27.28, outperforming the current state-of-the-art. The memory required exceeds a single GPU's capacity, making it difficult to render this scene at this quality without Grendel's techniques.

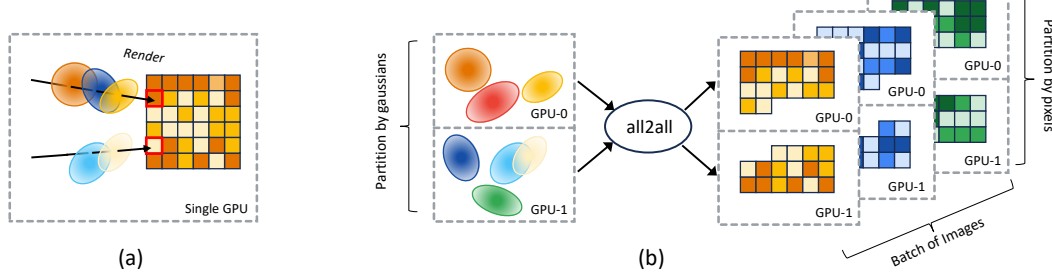

Figure 2: (a) Traditional 3DGS training pipeline using a single GPU vs. (b) Our Grendel system that distributes 3D Gaussians across multiple GPUs to alleviate the GPU memory bottleneck. We also partition the computation in the pixel and batch dimensions to for further speedup. Every square represents a $16 \times 16$ block of pixels.

## 2    GAUSSIAN SPLATTING: BACKGROUND, OPPORTUNITIES AND CHALLENGES

3D Gaussian Splatting (Kerbl et al., 2023) (3DGS) is a rendering method that represents 3D scenes using a (potentially large) set of anistropic 3D Gaussians. Each 3D Gaussian is represented by four learnable parameters: (a) its 3D position $x_i \in \mathbb{R}^3$; (b) its shape described by a 3D covariance matrix computed using the Guassian's scaling vector $s_i \in \mathbb{R}^3$ and rotation vector $q_i \in \mathbb{R}^4$; (c) its opacity $\alpha_i \in \mathbb{R}$; and (d) its spherical harmonics $sh_i \in \mathbb{R}^{48}$. The color contribution of each Gaussian is detemined by these parameters and by the viewing-direction.

### 2.1    BACKGROUND ON 3D GAUSSIAN TRAINING

To train 3DGS, the user provides an initial point cloud (may be random or estimated) for a scene and a set of posed images from different angles. The training process initializes Gaussians using the point cloud. Each training step selects a random camera view and uses the current Gaussian parameters to render the view. It then computes loss by comparing the rendered image to the ground truth, and uses back-propagation to update the Gaussian parameters. The training process also uses an adaptive densification mechanism to add Gaussians to under-reconstructed areas, by cloning or splitting existing ones based on their position variance and scale threshold, with more details in A.1.

Concretely, the training pipeline consists of four steps: Gaussian transformation, image rendering, loss calculation, and backpropagation. Standard approaches to backpropagation are used in this setting, and we detail the remaining three steps below:

1. **Gaussian transformation:** Given a camera view $v$ and the associated screen space, each Gaussian $i$ is transformed and projected to determine its position $x_{v,i} \in \mathbb{R}^2$ on screen, its distance $depth_{v,i} \in \mathbb{R}$ from the screen, and its coverage (or footprint radius) $radius_{v,i} \in \mathbb{R}$. Additionally, the color of each Gaussian $c_{v,i}$ is determined according to the viewing direction using its learnable spherical harmonics coefficients $sh_i \in \mathbb{R}^{48}$.
2. **Rendering:** After Gaussian transformation, the image is rendered by computing each pixel's color. To do so, for a given pixel $p$, 3DGS first finds all Gaussians that intersect with $p$. We say that a Gaussian $i$ intersects with $p$ if $p$ lies within $radius_{v,i}$ of the Gaussian $i$'s projected center $x_{v,i}$. Then 3DGS iterates over intersecting Gaussians in increasing depth (i.e. in increasing $depth_{v,i}$) and uses alpha-composition to combine their contributions until a threshold opacity has been reached.
3. **Loss calculation:** Finally, the 3DGS computes the L1 and SSIM loss by comparing the rendered image to the ground truth image. The L1 loss measures the absolute difference between pixel colors, while the SSIM loss measures the similarity between pixel windows. Both metrics are computed per-pixel for both forward and backward implementations.

### 2.2    OPPORTUNITIES AND CHALLENGES IN DISTRIBUTING 3DGS

In designing Grendel for scaling up 3D Gaussian Splatting training, we exploit the following opportunities in the above-described training process and address several challenges:

**Opportunity: mixed parallelism.** Each of the steps described above is inherently parallel but requires different kinds of work partitioning. In particular, the Gaussian transformation step operates on individual Gaussians and thus should be partitioned by Gaussians. On the other hand, the rendering and loss calculation steps operate on individual pixels(or pixels windows for SSIM loss) and thus should be partitioned by pixel.

**Opportunity: spatial locality.** Most Gaussians intersect a small contiguous area of the rendered image due to their typically small radius. As illustrated in Figure 3, 90% of the 3D Gaussians in three scenes (Rubble, Bicycle, and Train) have a radius $< 2\%$ of image width. Consequently, a pixel is affected by a small subset of the scene's 3D Gaussians, with significant overlap among neighboring pixels' Gaussians.

**Challenge: dynamic and unbalanced workloads.** Different image areas intersect varying quantities of Gaussians, as shown in Fig 4. For instance, an image region containing the sky likely corresponds to fewer Gaussians than a region with a person. Additionally, the density, position, shape, and opacity of Gaussians change throughout training. Therefore, the number of Gaussians and their mapping to pixels evolve over time, leading to computational workload imbalances across different image regions and over the training period. Fixed partitioning schemes thus suffer from load imbalance. Refer to the Appendix §A.4 for further details.

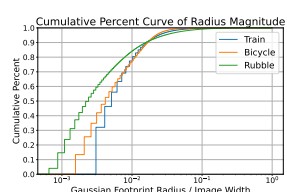

Figure 3: Cumulative percents of $Radius_i$ relative to image width

**Challenge: absence of batching.** Current 3DGS systems process images one at a time, which suffices for single GPU training. However, as shown in §5, this approach is inefficient in a distributed setting with multiple GPUs. Effective training with larger batch sizes necessitates an understanding of the unique optimization dynamics of 3DGS, which may differ from those of conventional neural networks.

## 3 SYSTEM DESIGN

Here, we describe how Grendel exploits the mixed parallelism and spatial locality of 3DGS (§3.1) to address the challenge of dynamic and unbalanced workloads (§3.2).

### 3.1 MIXED PARALLELISM TRAINING

Figure 2(b) provides an overview of Grendel's design. Grendel distributes work according to 3DGS' *mixed parallelism*: it uses Gaussian-wise distribution—where each GPU operates on a disjoint subset of Gaussians–for the Gaussian transformation step, and pixel-wise distribution–where each GPU operates on a disjoint subset of pixels— for the image rendering and loss computation step. The *spatial locality* characteristic allows Grendel to benefit from sparse all-to-all communication when transitioning between these stages.

**Gaussian-wise Distribution.** Grendel partitions the Gaussians, including their parameters and optimizer states, and distributes them uniformly across GPUs. Then, each GPU independently computes the Gaussian transformation for the set of 3D Gaussians assigned to it. We found that the amount of computation required does not significantly vary across Guassians, and thus evenly distributing Gaussians across GPUs allows us to fit the maximal number of Gaussians while speeding up computation linearly for this step.

**Pixel-wise Distribution.** We distribute contiguous image areas across GPUs for the image rendering and loss computation steps. Distributing contiguous areas allows us to exploit spatial locality and reduce the number of Gaussians transferred among GPUs. In our implementation, we partition each image in a batch by dividing it into $16 \times 16$-pixel blocks, serializing the blocks, and then distributing consecutive subsequences of blocks to different GPUs using an adaptive strategy (§3.2). For batching, each GPU can be assigned blocks from different images in a batch, as shown in Figure 2(b).

**Transferring Gaussians with sparse all-to-all communication.** To render an image pixel, a GPU needs access to Gaussians that intersect the pixel, which cannot be pre-determined as they are view-dependent and change during training. Therefore, Grendel includes a communication step after the Gaussian transformation. As 3DGS exhibits spatial locality, each pixel partition only requires a small subset of all 3D Gaussians. We leverage this to reduce communication: each GPU first decides the set of intersecting Gaussians for rendering a pixel partition (Figure 5) before using a sparse all-to-all communication to retrieve Gaussians intersecting with any pixels in the partition. A reversed all-to-all communication is done during the backward pass.

Although Grendel's design bears some resemblance to FSDP (Zhao et al., 2023) used for distributed neural network training, there are important differences. Firstly, unlike weight sharding in FSDP, Gaussian-wise distribution in Grendel is not merely for storage but for also for computation (the Gaussian transformation). Secondly, unlike FSDP which transfers weight shards using the dense all-gather communication, Grendel transfers only relevant Guassians using sparse all-to-all communication.

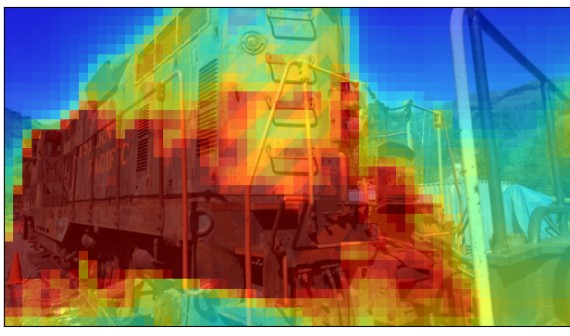 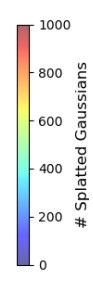 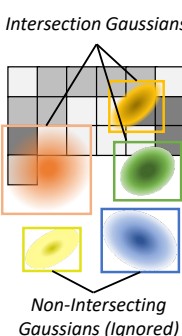

Figure 4: We present a heatmap of the per-tile imbalance in the number of rendered Gaussians on the Train Dataset (Knapitsch et al., 2017). Redder tiles indicate more Gaussian splats. Distant, low-detail areas like the sky need fewer Gaussians than detailed foreground regions like the train, highlighting an imbalance in rendering intensity.

Figure 5: Each GPU only considers Gaussians whose footprints intersect with its assigned pixel render area.

### 3.2 ITERATIVE WORKLOAD REBALANCING

**Pixel-wise Distribution Rebalancing.** As discussed in §2.2, the computational load of rendering a pixel varies across space (different pixels) and time (different training iterations). Thus, unlike in distributed neural network training, a uniform or fixed distribution cannot guarantee balanced workloads, so an adaptive pixel distribution strategy is needed.

We record the rendering time of each pixel of each training image during every epoch after the first few. Since the scene generally changes smoothly between consecutive epochs during training, the rendering time of each pixel also changes slowly. Therefore, the rendering times from previous epochs form a good estimate of a pixel's rendering time in the current epoch. Based on this estimate, we can adaptively assign pixels to different GPUs such that the workloads are approximately balanced.

Specifically, Grendel measures the running time (including image rendering, loss computation, and the corresponding backward computation) of each block of pixels assigned to a GPU, computes the average per-pixel computation time for the GPU, and uses this average to approximate the computation time for any pixel $p$ assigned to the GPU. For example, if a GPU is assigned pixels $p_0$ through $p_n$, and takes time $t$ for all of these pixels, then Grendel assumes that pixel $p_i$ where $i \in [0, n]$ requires $\frac{t}{n}$ time for computation. In subsequent iterations, the image is re-split so that the sum of the computation time for pixels assigned to all GPUs are equal. In our implementation, we use $16 \times 16$ pixel blocks as the split granularity. We show the pseudocode (Algorithm 1) for calculating the Division Points to split an image into load-balanced subsequences of blocks.

---

**Algorithm 1** Calculation of Division Points

**Require:** $B$(number of pixel blocks), $G$(number of GPUs), $ET_j$(Estimated runtime per pixel block)
**Ensure:** $DP$ (division points)
  1: $CT \leftarrow$ TORCH.CUMSUM$(ET)$                              ▷ Cumulative sum of ET
  2: $ET_{gpu} \leftarrow CT[B-1]/G$                          ▷ Estimated runtime per GPU
  3: $TH \leftarrow$ TORCH.ARANGE$(0, G) \cdot ET_{gpu}$            ▷ Thresholds for Division Points
  4: $DP \leftarrow$ TORCH.SEARCHSORTED$(CT, TH)$           ▷ Division Points
  5: **return** $DP$

---

**Gaussian-wise Distribution Rebalancing.** When training starts, we distribute 3D Gaussians uniformly among the GPUs. As training progresses, new Gaussians are added by cloning and splitting existing ones(§2.1). Newly added Gaussians make the distribution imbalanced as different Gaussians densify at different rates that depend on the scene's local details. Therefore, we redistribute the 3D Gaussians after every few densification steps to restore uniformity.

## 4 SCALING HYPERPARAMETERS FOR BATCHED TRAINING

To efficiently scale to multiple GPUs, Grendel increases the batch size beyond one, enabling partitioning of both images and pixels within each image, as shown in Figure 2(b).

However, increasing the batch size without adjusting hyperparameters, particularly the learning rate, can result in unstable and inefficient training (Goyal et al., 2017; Qiao et al., 2021), and hyperparameter tuning is often tedious. Though some methods simplify learning-rate tuning for deep neural networks, they either build on SGD (Goyal et al., 2017) (we use Adam) or they leverage

the layer-wise structure of neural networks (Ginsburg et al., 2018; You et al., 2020) (3DGS is not neural network). Our result is driven by the *Independent Gradients Hypothesis* for 3DGS training. Inspired by (Malladi et al., 2022), we derive a scaling rule for the hyperparameters of Adam, which suggests the same learning rate scaling as recent works (Malladi et al., 2022; Granziol et al., 2022) but a different scaling for $\beta_1$ and $\beta_2$ that works better for 3DGS.

We propose to scale Adam's **learning rate** and **momentum** based on batch size as follows:

$$\lambda' = \lambda \times \sqrt{\text{batch\_size}} \tag{1}$$

$$\beta_1', \beta_2' = \beta_1^{\text{batch\_size}}, \beta_2^{\text{batch\_size}} \tag{2}$$

where $\lambda$ is the original learning rate, and $\beta_1, \beta_2$ are the original first and second moments in Adam. $\lambda', \beta_1', \beta_2'$ are the adjusted hyperparameters to work with a greater batch size. We refer to these as the square-root learning rate scaling and the exponential momentum scaling rules.

**Independent Gradients Hypothesis.** To derive these scaling rules, we first consider 3D GS training in a simplified setting, assuming that gradients calculated from each camera view are *independent* of those induced from other views. Consequently, if we are given a batch of $b$ camera views, taking $b$ sequential gradient descent steps for each view in the batch is equivalent to taking one bigger step where the gradients are summed together. If we were using the vanilla gradient descent algorithm and averaging the gradients in a batch, setting the learning rate to scale linearly with the batch size achieves this equivalence. However, 3D GS uses Adam, an adaptive learning rate optimizer that (1) divides the gradients by the square root of the per-parameter second moment estimate, and (2) uses momentum to combine current gradients and past gradients in an exponential-moving-average fashion, making a bigger update different from simply summing up smaller batch-size-one updates. Under the *independent gradients hypothesis*, we derive the following corrections to Adam hyperparameters to approximate batch-size-one training with a larger batch:

Let us denote $g_k$ as the gradient of some parameter evaluated at view $k$, and $g = \frac{\sum_{j \in V} g_j}{|V|}$ as the full-batch gradient (mean of gradients across views), where $V$ is the set of all views. Let us further assume $\mathbb{E}[g_k] = 0$ for all $k$. By the independence assumption: $\text{Cov}(g_k, g_j) = \mathbb{E}[(g_k - 0)(g_j - 0)] = 0$ when $k \neq j$ and $\mathbb{E}[(g_k)^2]$ when $k = j$.

Then, parameter update from a batch-size-1 Adam step (without momentum) on view $k$ is:

$$\Delta^{\{k\}} = \frac{g_k}{\sqrt{\mathbb{E}\left[\mathbb{E}_{j \in V}\left[g_j^2\right]\right]}} = \frac{g_k}{\sqrt{\mathbb{E}\left[|V|g^2\right]}} = \frac{g_k}{\sqrt{|V|}\sqrt{\mathbb{E}\left[g^2\right]}}.$$

However, the parameter update from one Adam step (without momentum) on a batch of views $B \subseteq V$ of size $b$ is:

$$\Delta^{\{B\}} = \frac{\sum_{k \in B} g_k/b}{\sqrt{\mathbb{E}\left[\mathbb{E}_{B' \subseteq V}\left[\left(\sum_{j \in B'} g_j/b\right)^2\right]\right]}} = \frac{\sum_{k \in B} g_k/b}{\sqrt{\mathbb{E}\left[\frac{|V|}{b}g^2\right]}} = \frac{\sum_{k \in B} g_k/b}{\sqrt{\frac{|V|}{b}}\sqrt{\mathbb{E}\left[g^2\right]}} = \frac{1}{\sqrt{b}}\frac{\sum_{k \in B} g_k}{\sqrt{|V|}\sqrt{\mathbb{E}\left[g^2\right]}}.$$

Thus, setting the learning rate $\lambda' = \lambda \times \sqrt{b}$ allows the batch update $\Delta^{\{B\}}$ to match with the total individual updates $\sum_{k \in B} \Delta^{\{k\}}$. Alongside the square-root learning rate scaling (Eq 1), we also propose an exponential momentum scaling to accommodate larger batches (Eq 2). Initially used by Busbridge et al. (2023), this rule scales the momentum parameters with $\beta' = \beta^{\text{batch\_size}}$, which exponentially decreases the influence of past gradients when the batch size increases.

We wish to stress that in the real world, even though some cameras share similar poses, a set of random cameras generally observe different parts of a scene, hence the gradients in a batch are mostly sparse and can be thought of as roughly independent. We empirically study the independent gradient hypothesis and evaluate our proposed scaling rules.

### 4.1 EMPIRICAL EVIDENCE OF *Independent Gradients*

To see if the *Independent Gradients Hypothesis* holds in practice, we analyze the average per-parameter variance of the gradients in real-world settings. We plot the sparsity and variance of the gradients of the diffuse color parameters starting at pre-trained checkpoints on the "Rubble" dataset (Turki et al., 2022) against the batch size in Figure 6. We find that the inverse of the variance increases roughly linearly, then transitions into a plateau. We find this behavior in all three checkpoint iterations, representing early, middle, and late training stages. The initial linear increase of the precision suggests that gradients are roughly uncorrelated at batch sizes used in this work (up to 32) and supports the

*independent gradients hypothesis*. While a single image may have sparse gradients; in a large batch, gradients overlap and become less sparse. They also grow more correlated, as camera with similar poses are expected to offer similar gradients.

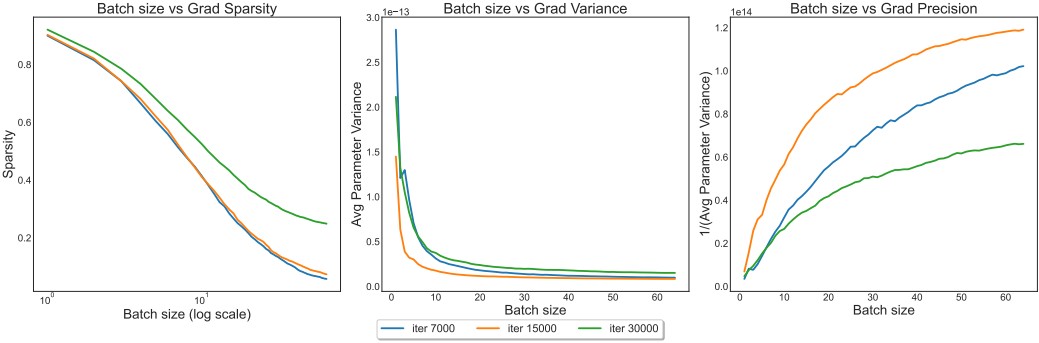

Figure 6: Gradients are roughly uncorrelated in practice. On the "Rubble" dataset (Turki et al., 2022), the inverse of the average parameter variance increases linearly, then rises to a plateau, suggesting that the gradients are roughly uncorrelated initially but become less so as the batch size becomes large. Averaged over 32 random trials.

### 4.2 EMPIRICAL TESTING OF PROPOSED SCALING RULES

To empirically validate the proposed learning rate and momentum scaling rules, we train the "Rubble" scene up to iteration 15,000 using a batch size of 1. Then, we reset the Adam optimizer states and continue training with different batch sizes. We compare how well different learning rate and momentum scaling rules maintain a similar training trajectory when switching to larger batch sizes in Figure 12(Appendix B.3). Without loss of generality, we focus on the diffuse color parameters for this analysis. Figure 12a compares three different learning rate scaling rules ∈ [constant, sqrt, linear] where only our proposed "sqrt" holds a high update cosine similarity and a similar update magnitude across different training batch sizes. Similarly, 12b shows our proposed exponential momentum scaling rule keeps update cosine similarity higher than the alternative which leaves the momentum coefficients unchanged.

## 5 EVALUATION

Our evaluation aims to demonstrate Grendel's scalability, showing both that it can render high-resolution images from large scenes, and that its performance scales with additional hardware resources. We also compare Grendel's system-level parallelization with CityGaussian's (Liu et al., 2024) divide-and-conquer approach in 5.3. The ablation study on dynamic load balancing and learning rate scaling strategies is presented in Appendix C.1.

### 5.1 SETTING AND DATASETS

**Experimental Setup.** We conducted our evaluation in the Perlmutter GPU cluster NERSC. Each node we used was equipped with 4 A100 GPUs with 40GB of GPU memory, and interconnected with each other using 25GB/s NVLink per direction. Servers were connected to each other using a 200Gbps Slingshot network.

**Datasets.** We evaluate Grendel using the datasets and corresponding resolution settings shown in Table 1. Of these, Rubble and MatrixCity Block_All represent the large scale datasets that are out of reach for most existing 3DGS systems, while other datasets are commonly used in 3DGS papers. These datasets vary in area size and resolution to comprehensively test our system.

**Evaluation Metrics.** We report image quality using SSIM, PSNR and LPIPS values, and throughput in training images per second. We take both forward and backward time into consideration of throughput. And note that throughput in images per second may differ from throughput in iterations per second, as one iteration includes the batch size number of images.

### 5.2 PERFORMANCE AND MEMORY SCALING

We start by evaluating Grendel's scaling, and how additional GPUs impact computation performance and memory.

| Dataset | # Scenes | Resolutions | # Images | Test Set Setting |
|---|---|---|---|---|
| Tanks & Temple (Knapitsch et al., 2017) | 2 | $\sim 1K$ | 251 to 301 | 1/8 of all images |
| DeepBlending (Hedman et al., 2018) | 2 | $\sim 1K$ | 225 to 263 | 1/8 of all images |
| Mip-NeRF 360 (Barron et al., 2022) | 9 | 1080P | 100 to 330 | 1/8 of all images |
| Rubble (Turki et al., 2022) | 1 | $4591 \times 3436$ | 1657 | official test set |
| MatrixCity Block_All (Li et al., 2023) | 1 | 1080P | 5620 | official test set |

Table 1: Scenes used in our evaluation: We cover scenes of varying sizes and resolutions.

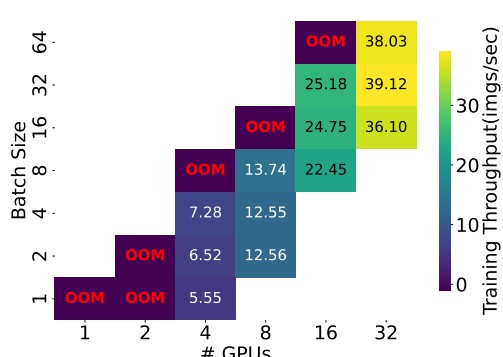

Figure 7: To avoid OOM, 4 GPUs are needed to train the large 4K "Rubble" scene. We further improve throughput by distributing across even more GPUs and increasing the batch size.

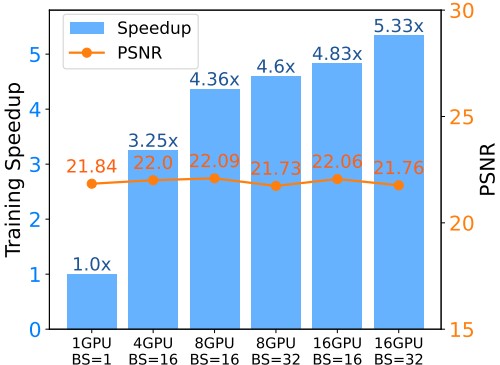

Figure 8: Even for the small 'Train' scene, our distributed training with larger batch sizes achieves speedup without compromising test PSNR. All configs train for 30K total images.

**Computation.** We evaluated how additional GPUs impact Grendel's performance using both large-scale (Rubble) and small-scale (Train and Mip-Nerf360) datasets.

We used the Rubble scene to evaluate the training throughput. For this experiment we used 35 million Gaussians which have been trained to convergence. Because of the time required to render 4K images for this scene, we measured throughput for training over another 10,000 images times, and in Figure 7 we report throughput (in images per second) as we vary the number of GPUs (x-axis) and batch size (y-axis). We observe that we cannot render this scene with a single GPU (regardless of batch size) because of its memory requirements. Furthermore, both increasing the number of GPUs and increasing batch size yield performance improvements: performance increases from 5.55 images per second (4 GPUs, batch size 1) to 38.03 images per-second (32 GPUs, batch size 64).

Next, we use the $980 \times 545$ resolution Train scene to evaluate both throughput and image quality during scaling. Due to its small, low-resolution nature, it can be trained from scratch for each experiment. Our results in Figure 8 show that additional GPUs improve throughput while maintaining image quality when trained with the same total number of images. Notably, our 16-GPU setup with a batch size of 32 completes training on 30K images in just 2 minutes and 42.97 seconds, representing the state-of-the-art training speed to the best of our knowledge.

As shown in Figure 9, we also achieve a 3x to 4x speed up using 4 GPU and a batch size of 4, without PSNR degradation across 13 scenes from the Mip-Nerf360 dataset (first half) and the Tanks & Temple and Deep Blending datasets (second half). We use default hyperparameters from the 3DGS repository (Kerbl et al., 2023). We train on the same number of images: 50k for Mip-NeRF 360 and 30k for the slightly smaller TT & DB datasets, to ensure convergence and a fair comparison.

**Memory Scaling.** Scaling the number of GPUs increases memory, allowing more Gaussians to represent a scene. We tested this by adding Gaussians through densification until we ran out of memory. Figure 14 (In Appendix C.3) illustrates the number of Gaussians that Grendel can accommodate (in millions) with batch sizes of 1, 4, and 16, as the number of GPUs increases. The results demonstrate linear scaling. In §5.2.1 we show the utility of using additional Gaussians.

We provide additional details about experiments from this section in Appendix C.3.

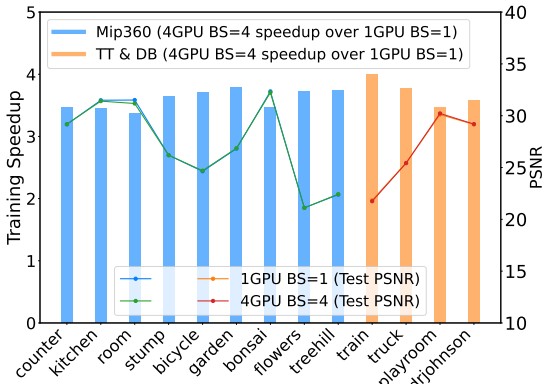

Figure 9: Training Speedup and PSNR on Mip-NeRF360 and Tanks&Temples+Deep Blending.

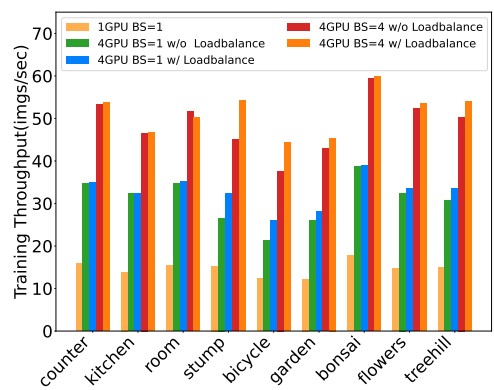

Figure 10: Speedup from Iterative Load Balancing and increased batch sizes on Mip-NeRF360.

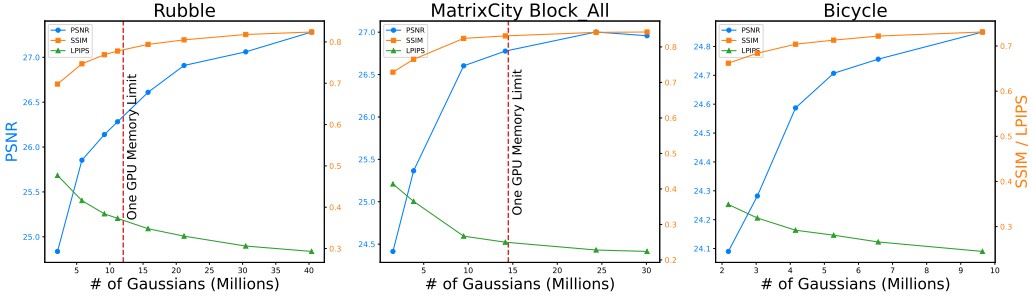

Figure 11: Scalability Statistics: Gaussian Quantity vs. Reconstruction Quality
Using more Gaussians results in better test metrics for reconstruction. The red line indicates the number of Gaussians a single GPU can handle, which is insufficient for achieving high-quality results.

### 5.2.1 GAUSSIAN QUANTITY VS. RECONSTRUCTION QUALITY

Scaling to multiple GPUs allows Grendel to use a larger number of Gaussians to represent scenes. A larger number of Gaussians can capture fine grained scene details, and should thus be able to better reconstruct large-scale, high-resolution scenes. We evaluated this effect using three scenes: Rubble, MatrixCity Block_All, and Bicycle, and varied the number of Gaussians used by changing densification settings: we lowered the gradient norm threshold to initiate densification and reduced the threshold for splitting Gaussians instead of cloning until the densification mechanism produced the target number of Gaussians without manual interference. We rendered Rubble and Matrix City Block_All using 16 GPUs and a batch size of 16, while we used 4 GPUs and a batch size of 4 for Bicycle. The difference in number of GPUs and batch sizes is due to differences in scene sizes: bicycle is much smaller than the other two datasets.

In Figure 11, we show that image quality metrics (PSNR, SSIM and LPIPS) improve as we add more Gaussians. The red line in the Rubble and Matrix City Block_All graphs shows the number of Gaussians that can fit on a single GPU (Bicycle, being smaller, can be rendered on a single GPU). Figure 13 shows the rendered images as we scale Gaussians quantity, and demonstrates the quality improvements are human visible. These results demonstrate benefits of using more Gaussians, and demonstrate the necessity of multi-GPU 3DGS training systems like Grendel.

### 5.3 COMPARISON TO CITYGAUSSIAN

We evaluate Grendel's system-level parallelization against CityGaussian(Liu et al., 2024), the open-source 3DGS divide-and-conquer solution. Since CityGaussian is not designed for high-resolution reconstruction, we perform these experiments on downsampled scenes: Rubble (4x downsampled) (Turki et al., 2022), Building (4x downsampled)(Turki et al., 2022), and MatrixCity Block_All (downsampled to a width of 1600 pixels)(Li et al., 2023). All experiments are conducted on 4 A100s. We provide two comparison points for CityGaussian. *CityGS Official* uses the authors'

official script [1], yielding PSNR numbers comparable to those reported in (Liu et al., 2024). *CityGS Official* experiments train Rubble, Building, and MatrixCity Block_All on a total of 300000, 630000, and 1110000 images, respectively. *CityGS 200K-images* experiments use the same configuration as *CityGS Official*, except for training each scene with a total of only 200000 images, in order to compare the convergence rate with Grendel. In *Grendel 200K-images*, each scene is trained using Grendel for a total of 200000 images, matching the count in *CityGS 200K-images*.

| Method | Rubble | | | | Building | | | | MatrixCity Block_All | | | |
|---|---|---|---|---|---|---|---|---|---|---|---|---|
| | PSNR | SSIM | LPIPS | Total Time | PSNR | SSIM | LPIPS | Total Time | PSNR | SSIM | LPIPS | Total Time |
| CityGS Official | 25.88 | 0.813 | 0.231 | 2.88 hrs | 22.14 | **0.784** | **0.241** | 4.57 hrs | **27.41** | **0.864** | **0.205** | 8.25 hrs |
| CityGS 200K-images | 25.40 | 0.796 | 0.249 | 2.18 hrs | 20.32 | 0.725 | 0.299 | 2.22 hrs | 23.68 | 0.701 | 0.422 | 3.60 hrs |
| Grendel 200K-images (ours) | **27.39** | **0.859** | **0.195** | **0.85 hrs** | **22.69** | 0.778 | 0.242 | **0.90 hrs** | 27.33 | 0.859 | **0.205** | **1.22 hrs** |

Table 2: Quantitative evaluation of Grendel compared to CityGaussian(Liu et al., 2024). We report PSNR↑, SSIM↑, and LPIPS↓ on test views, along with Total Training Time. The **best** and second best results are highlighted. Time Decomposition is provided in Table 10(In Appendix 5).

As shown in Table 2, Grendel achieves test PSNR comparable to, or surpassing those of *CityGS Official*. Grendel is much faster—achieving 3x-6.7x speed improvements over *CityGS Official*. Compared to *CityGS 200K-images*, Grendel demonstrates greater training efficiency in both convergence rate and training time. Our experience also shows that Grendel is simpler to use. CityGS requires running several separate procedures each of which requires hyperparameter tuning. By contrast, running Grendel over multiple GPUs requires similar efforts as the original 3DGS.

## 6    RELATED WORKS

*Large-scale scene reconstruction.* Prior works have proposed the divide-and-conquer approach to scale 3DGS to work with large scenes. VastGaussian (Lin et al., 2024), CityGaussian (Liu et al., 2024), and Hierarchical Gaussian (Kerbl et al., 2024) divide large scenes into small regions and train each region separately, and then merge the resulting sub-models. Hierarchical Gaussian (Kerbl et al., 2024), Octree-GS (Ren et al., 2024) and CityGaussisan (Liu et al., 2024) describe level-of-detail based approaches to adaptively reduce the number of Gaussians considered for distant objects. Grendel's system-level parallelization is complementary to these algorithmic innovations. For example, the initial coarse training step in both CityGaussisan (Liu et al., 2024), and Hierarchical Gaussian (Kerbl et al., 2024) can utilize Grendel for multi-GPU acceleration. Similar methods (Turki et al., 2022; Yuanbo et al., 2022; Li et al., 2024b) have been employed to scale NeRF, but they are not directly applicable to 3DGS due to its distinct computation pattern, as discussed in Section 1.

*Distributed Training for 3DGS.* DOGS (Chen & Lee, 2024) modifies training with ADMM distributed optimization, averaging Gaussians shared across partition boundaries every 100 iterations. This method enables asynchronous training, which can potentially impact convergence rate. In contrast, Grendel preserves the original 3DGS algorithm, maintaining the same convergence characteristics as the single-GPU version. RetinaGS (Li et al., 2024a) also retains the original 3DGS algorithm but employs a distinct parallelization strategy. In RetinaGS, each GPU renders the entire image using its local partition of Gaussians, with the rendered outputs subsequently merged. However, this approach results in redundant computations, as many GPUs render pixels beyond the opacity saturation depth unnecessarily. Distributed training for neural networks is further discussed in Appendix D.

*Large Batch Size Training.* Large batch training has been widely adopted to improve the ML training performance and efficiency, but it has also been recognized by prior work (Keskar et al., 2017) that increasing batch size can adversely impact model performance. This has led to the development of empirical rules for neural networks training, including linear scaling and learning rate warmum for SGD (Goyal et al., 2017), square root scaling for Adam (Malladi et al., 2022) and layer-wised adaptive rate scaling (You et al., 2020; Ginsburg et al., 2018). The scaling rules in our work are inspired by these, but focuses on batch size scaling for 3DGS.

---

[1]https://github.com/DekuLiuTesla/CityGaussian/tree/main/scripts

## REPRODUCIBILITY STATEMENT

We provide hyperparameter details in Appendix C.3. We will also release code for training, rendering, and evaluation. We will release pre-trained model checkpoints for Rubble and MatrixCity datasets to reproduce the results reported in the paper.

## ACKNOWLEDGEMENTS

We thank Xichen Pan and Youming Deng for their help on paper writing. We thank Matthias Niessner for his insightful and constructive feedback on our manuscript. We thank Yixuan Li and Lihan Jiang from the MatrixCity team for their assistance in providing initial data points of their dataset. We thank Kaifeng Lyu for discussions on Adam training dynamics analysis. This research used resources of the National Energy Research Scientific Computing Center (NERSC), a U.S. Department of Energy Office of Science User Facility located at Lawrence Berkeley National Laboratory, operated under Contract No. DE-AC02-05CH11231.

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

## A    ADDITIONAL PRELIMINARIES & OBSERVATIONS DETAILS

This appendix provides additional information about 3DGS, beyond what was covered in §2.

### A.1    DENSIFICATION PROCESS

Densification is the process by which 3DGS adds more Gaussians to improve details in a particular region. A Gaussian that shows significant position variance across training steps, might either be clones or split. The decision on whether to clone or split depends on whether their scale exceeds a threshold. Hyperparameters determine the start and stop iteration for densification, its frequency, the gradient threshold for initiating densification, and the scale threshold that determines whether to split or clone. To create more Gaussians, we need to increase the stop iteration and frequency, and decrease the gradient threshold for densification. If we aim to capture more details using smaller Gaussians, we should lower the scale threshold to split more Gaussians. The training process also includes pruning strategies such as eliminating Gaussians with low opacity and using opacity reset techniques to remove redundant Gaussians.

### A.2    Z-BUFFER

The indices of intersecting gaussians for each pixel are stored in a Z-buffer, used in both forward and backward. This Z-buffer is the switch between View-dependent Gaussian Transformation and Pixel Render. Since a single gaussian can project onto multiple pixels within its footprint, the total size of all pixels' Z-buffers exceeds both the count of 3DGS and pixels. The Z-buffer itself, along with auxiliary buffers needed for sorting it, etc, consumes significant activation memory. This can also lead to out-of-memory (OOM) errors if the resolution, scene size, or batch size is increased.

### A.3    MIXED PARALLELISM

In the main text, some steps of 3DGS are not mentioned, but these steps can also be parallelized. The Gaussian transformation backward and gradient updates by the optimizer are also Gaussian-wise computations and will be distributed the same way as the Gaussian transformation forward. Similarly, the Render Backward and Loss Backward computations are pixel-wise and will be distributed just like the Render Forward.

Regarding the memory aspect, each Gaussian has independent transformed states, gradients, optimizer states, and parameters. Therefore, we save these states together on the corresponding GPU that contains their parameters. And activation states like significant Z-buffers, auxiliary buffers for sorting and other functions, loss intermediate activations are managed pixel-wise along with the image distribution.

Regarding densification mechanism, since we clone, split or prune Gaussians independently based on their variance, we perform this process locally on the GPU that stores them.

### A.4    DYNAMIC UNBALANCED WORKLOADS

Physical scenes are naturally sparse on a global scale. Different areas have different densities of 3D gaussians (i.e sky and a tree). Thus, the intensity of rendering not only varies from pixel to pixel within an image but also differs between various images, leading to workloads unbalance. Figure 4 shows the differences in render intensity across the image.

Besides, during the training, gaussians parameters are continuously changing. More precisely, the change of 3D position parameters and co-variance parameters affect each gaussian's coverage of pixels on the screen. The change of opacity parameters affect the number of gaussians that contribute to each pixel. Both of them lead to render intensity change. The densification process targets areas under construction. During training, simpler scene elements are completed first, allowing more complex parts to be progressively densified. This means Gaussians from different regions densify at varying rates. The dynamic nature of the workloads is more pronounced at the beginning of training, as it initially focuses on constructing the global structure before filling in local details.

The different computational steps have distinct characteristics in terms of workload dynamicity. Even though, the rendering computation is dynamic and unbalanced; computation intensity for loss calculation remains consistent across pixels, and the view-dependent transformation maintains a uniform computational intensity across gaussians. Actually, render forward and backward have different patterns of unbalance and dynamicity. The computational complexity for the forward process scales with the number of 3DGS intersecting the ray. In contrast, the complexity of the backward process depends on Gaussians that contributed to color and loss before reaching opacity saturation, typically those on the first surface. Then, running time for render forward and backward, loss forward and backward have different dominating influence factors, and every step takes a significant amount of time.

## B  ADDITIONAL DESIGN DETAILS

### B.1  SCHEDULING GRANULARITY: PIXEL BLOCK SIZE

In our design, we organize these pixels from all the images in a batch into a single row. Then, we divide this row into parts, and each GPU takes care of one part. However, if there are a lot of pixels, the strategy scheduler computation overhead will be very large. So we group the pixels into blocks of 16 by 16 pixels, put these blocks in a row and allocate these blocks instead. The size of block is essentially the scheduling granularity, which is a trade-off between scheduler overhead and uneven workloads due to additional blocks. After scheduling, we will have a 2D boolean array, compute_locally[i][j], indicating whether the pixel block at i-th row and j-th column should be computed by the local GPU. We will then render only the pixels within the blocks where compute_locally is true.

### B.2  GAUSSIAN DISTRIBUTION REBALANCE

An important observation is that distributing pixels to balance runtime doesn't necessarily balance the number of Gaussians each GPU touches in rendering; So, to minimize total communication volume, GPUs may need to store varying quantity of Gaussians based on the formula above. Specifically, only the forward runtime correlates directly with the number of touched 3DGS; however, the time it takes for pixel-wise loss calculations and rendering backward depends on the quantity of pixels and the count of gaussians that are indeed contributed to the rendered pixel color, respectively. In our experiments, random redistribution leads to fastest training here, even if its overall communication volume is not the minimum solution. Because in our experiment setting, we use NCCL all2all as the underlying communication primitive, which prefers the uniform send and receive volume among different GPU. If we change to use communication primitive that only cares about the total communication volume, then we may need to change to other redistribution strategy.

### B.3  SCALING RULE HYPERPARAMETER ABLATION

The effectiveness of our automatic hyperparameter scaling rules is demonstrated in the ablation study, as shown in Figure 12.

## C  ADDITIONAL EXPERIMENTS SETTING AND STATISTICS

### C.1  ABLATION STUDY

Figure 10 illustrates that our load balancing techniques and increased batch size significantly improve training throughput on 1080p Mip-NeRF360 dataset, compared to the one GPU baseline and our straightforward distributed system with a conventional batch size of one and no load balancing. Similar results are observed with the 4K Rubble Dataset, as shown in Figure 15. Although good speed can be achieved without load balancing, load balancing allows us to consistently achieve even higher throughput across various types and scales of scenes. The ablation study for the learning rate scaling strategies have already been discussed in 4.2, along with our analysis.

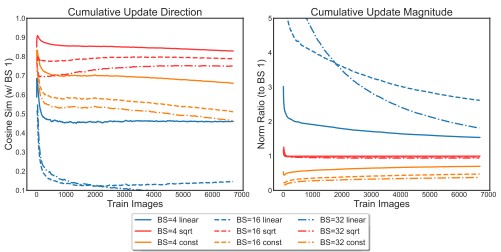 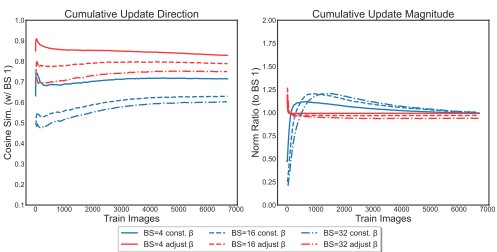

(a) Learning rate scaling rules vs. BS invariance.        (b) Momentum scaling rules vs. BS invariance.

Figure 12: We plot the training trajectories of the diffuse color parameters on "Rubble", when training with batch size $\in [4, 16, 32]$ using different learning rate and momentum scaling strategies. Cumulative weight updates using the square-root learning rate scaling rule (a, red curves) and exponential momentum scaling rule (b, red curves) maintain high cosine similarity to batch-size 1 updates and have norms that are roughly invariant to the batch size. All trajectories in (a) employ our proposed exponential momentum scaling but differing learning rate scaling; while all trajectories in (b) employ our proposed square-root learning rate scaling but differing momentum scaling.

## C.2   STATISTICS FOR MIP-NERF 360, TANK&TEMPLES AND DEEPBLENDING DATASETS

We provide full statistics of training results on Mip-NeRF 360, Tank&Temples and DeepBlending datasets in Table 3.

| Dataset | Scene | 1 GPU (bsz=1) | | 4 GPU (bsz=4) | |
|---|---|---|---|---|---|
| | | PSNR | Throughput | PSNR | Throughput |
| Mip-NeRF360 | counter | 29.16 | 16.25 | 29.19 | 56.24 |
| | kitchen | 31.49 | 14.24 | 31.40 | 49.16 |
| | room | 31.51 | 15.82 | 31.18 | 53.36 |
| | stump | 26.19 | 14.95 | 26.19 | 54.53 |
| | bicycle | 24.63 | 12.01 | 24.69 | 44.44 |
| | garden | 26.82 | 12.10 | 26.86 | 45.83 |
| | bonsai | 32.34 | 17.87 | 32.23 | 61.88 |
| | flowers | 21.11 | 14.47 | 21.10 | 53.94 |
| | treehill | 22.38 | 14.78 | 22.43 | 55.31 |
| Tank&Temples | train | 21.84 | 34.72 | 21.75 | 101.69 |
| | truck | 25.44 | 27.55 | 25.42 | 95.85 |
| DeepBlending | playroom | 30.11 | 21.98 | 30.22 | 75.38 |
| | drjohnson | 29.15 | 17.74 | 29.19 | 62.11 |

Table 3: Performance Comparison Between Non-Distribution and 4 GPU Distribution

## C.3   SCALABILITY

Table 4, 5 and 6 show the increased reconstruction quality with more gaussians. While many hyperparameters influence the number of Gaussians created by densification, we focused on adjusting three key parameters: (1) the stop iteration for densification, (2) the threshold for initiating densification, and (3) the threshold for deciding whether to split or clone a Gaussian. Initially, we gradually increased the densification stop iteration to 5,000 iterations. However, due to the pruning mechanism, this adjustment alone proved insufficient. Consequently, we also lowered the two thresholds to generate more Gaussians. For a fair comparison, all other densification parameters—such as the interval, start iteration, and opacity reset interval—were kept constant. For the Rubble scene, each experiment run for the same 125 epochs, exposing models to 200,000 images, ensuring consistency. Although training larger models for longer durations and lowering the positional learning rate improved results in my observations, we maintained consistent training steps and learning rates across all experiments to ensure fairness.

Table 7, 8 show the Throughput Scalability by Increasing batch size and leveraging more GPUs, for Rubble and Train scene, respectively. Essentially, more GPUs and larger batch size give higher throughput. More GPUs provide more computational power while larger batch size can utilze these GPUs better.

Table 9 demonstrates that additional GPUs increase available memory for more Gaussians, evaluated on the Rubble scene with various batch sizes reflecting different levels of activation memory usage. We can achieve linear scaling as illustrated in Figure 14. Essentially, more GPUs provide additional memory to store Gaussians, while a larger batch size increases activation memory usage, leaving less memory available for Gaussians.

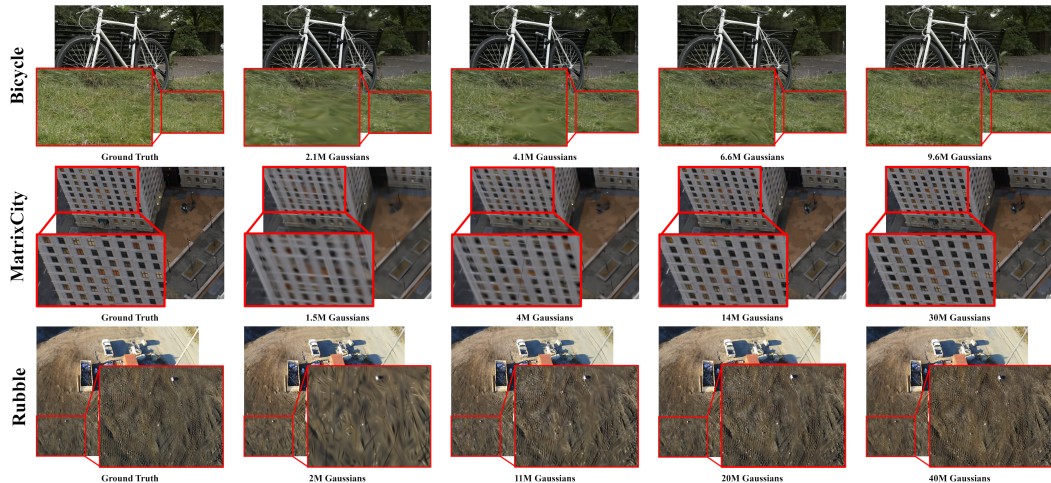

Figure 13: Visualization: Gaussian Quantity vs. Reconstruction Quality

| Experiment | n3dgs | Results | | | Densification Settings | |
|---|---|---|---|---|---|---|
| | | PSNR | SSIM | LPIPS | Stop Iter | Thresholds |
| EXP 1 | 2114045 | 24.84 | 0.70 | 0.48 | 5000 | (0.0002, 0.01) |
| EXP 2 | 5793396 | 25.85 | 0.75 | 0.42 | 15000 | (0.0002, 0.01) |
| EXP 3 | 9173931 | 26.14 | 0.77 | 0.38 | 50000 | (0.0002, 0.01) |
| EXP 4 | 11168630 | 26.28 | 0.78 | 0.37 | 50000 | (0.00018, 0.008) |
| EXP 5 | 15754744 | 26.61 | 0.79 | 0.35 | 50000 | (0.00015, 0.005) |
| EXP 6 | 21177774 | 26.91 | 0.80 | 0.33 | 50000 | (0.00013, 0.003) |
| EXP 7 | 30474202 | 27.06 | 0.82 | 0.31 | 50000 | (0.0001, 0.002) |
| EXP 8 | 40397406 | **27.28** | **0.82** | **0.29** | 50000 | (0.00008, 0.0016) |

Table 4: Scalablity on Rubble: Gaussian Quantity, Results and Hyperparameter Settings

| Experiment | n3dgs | Results | | | Densification Settings | |
|---|---|---|---|---|---|---|
| | | PSNR | SSIM | LPIPS | # Start Points | # Densify Iter |
| EXP 1 | 1545568 | 24.41 | 0.73 | 0.41 | 1545568 | 0 |
| EXP 2 | 3867136 | 25.36 | 0.77 | 0.36 | 3867136 | 0 |
| EXP 3 | 9485755 | 26.6 | 0.82 | 0.27 | 7743616 | 5000 |
| EXP 4 | 14165332 | 26.78 | 0.83 | 0.25 | 15540941 | 5000 |
| EXP 5 | 24355726 | **27.0** | **0.84** | 0.23 | 15540941 | 30000 |
| EXP 6 | 30074630 | 26.96 | **0.84** | **0.22** | 15540941 | 40000 |

Table 5: MatrixCity Block_All Statistics: Gaussian Quantity, Results and Hyperparameter Settings

| Experiment | n3dgs | Results | | | Densification Settings | |
|---|---|---|---|---|---|---|
| | | PSNR | SSIM | LPIPS | Stop Iter | Thresholds |
| EXP 1 | 2185112 | 24.09 | 0.66 | 0.35 | 5000 | (0.0002, 0.01) |
| EXP 2 | 3035508 | 24.28 | 0.68 | 0.32 | 7000 | (0.0002, 0.01) |
| EXP 3 | 4154806 | 24.59 | 0.70 | 0.29 | 10000 | (0.0002, 0.01) |
| EXP 4 | 5272686 | 24.71 | 0.71 | 0.28 | 15000 | (0.0002, 0.01) |
| EXP 5 | 6579244 | 24.76 | 0.72 | 0.27 | 15000 | (0.00018, 0.008) |
| EXP 6 | 9636072 | **24.85** | **0.73** | **0.25** | 15000 | (0.00015, 0.005) |

Table 6: Bicycle Statistics: Gaussian Quantity, Results and Hyperparameter settings

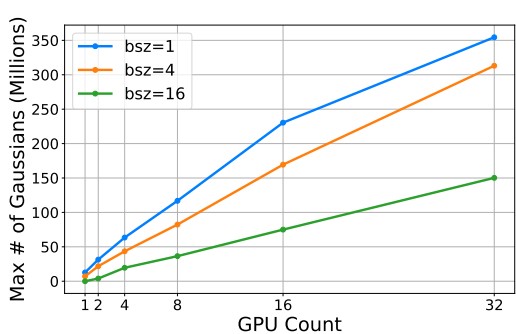

Figure 14: More GPUs provide additional memory to support more Gaussians before encountering OOM.

Figure 15: Load balancing and larger batch accelerate training on 4K Rubble Scene

### C.4 RENDER SPEEDUP

We compare Grendel's rendering speeds on single-GPU and 4-GPU setups for the Rubble, MatrixCity Block-All, and Bicycle scenes, using our state-of-the-art trained Gaussian models corresponding to Figure 11. We experiment with various resolutions and scene scales, rendering one image at a time for fair comparison. The 4-GPU setup achieves a speedup of 1.88x to 2.63x, as shown in Table 11.

## D ADDITIONAL RELATED WORKS

*Distributed training for neural networks.* Existing work have exploited various types of parallelism to train neural networks across GPUs. These include data parallelism (Li et al., 2020), tensor parallelism (Shoeybi et al., 2020; Narayanan et al., 2021), pipeline parallelism (Huang et al., 2019; Narayanan et al., 2019) and FSDP (Zhao et al., 2023; Rajbhandari et al., 2020). Several systems also support multiple types parallelism and/or aim to automatically partition the workload to optimize speed (Zheng et al., 2022; Xu et al., 2021; Wang et al., 2019). However, as we discussed earlier (§1), neural network and 3DGS have very different computation patterns. The former performs repeated layer-wise computation dominated by dense matrix multiply operations while the latter's 3 stages

| GPU Count | bsz=1 | bsz=2 | bsz=4 | bsz=8 | bsz=16 | bsz=32 | bsz=64 |
|---|---|---|---|---|---|---|---|
| 1 GPU | OOM | | | | | | |
| 2 GPU | OOM | | | | | | |
| 4 GPU | 5.55 | 6.52 | 7.28 | OOM | | | |
| 8 GPU | | 12.56 | 12.55 | 13.74 | OOM | | |
| 16 GPU | | | | 22.45 | 24.75 | 25.18 | OOM |
| 32 GPU | | | | | 36.10 | 39.12 | 38.03 |

Table 7: Scalability on Rubble: Speed up from More GPU and Larger Batch Size

| Experiment | # GPU | Batch Size | Throughput | PSNR |
|------------|-------|------------|------------|------|
| EXP 1 | 1 | 1 | 34.72 | 21.84 |
| EXP 2 | 4 | 16 | 112.78 | 22.01 |
| EXP 3 | 8 | 16 | 151.52 | 22.09 |
| EXP 4 | 8 | 32 | 159.57 | 21.73 |
| EXP 5 | 16 | 16 | 167.60 | 22.06 |
| EXP 6 | 16 | 32 | 185.19 | 21.76 |

Table 8: Scalability on Train: Speed up from More GPU and Larger Batch Size

| GPU Count | bsz=1 | bsz=4 | bsz=16 |
|-----------|-------|-------|--------|
| 1 GPU | 12.71 M | 7.10 M | OOM |
| 2 GPU | 31.40 M | 21.80 M | 3.91 M |
| 4 GPU | 63.44 M | 43.48 M | 19.55 M |
| 8 GPU | 116.85 M | 82.31 M | 36.44 M |
| 16 GPU | 230.41 M | 169.37 M | 74.98 M |
| 32 GPU | 354.46 M | 313.10 M | 150.21 M |

Table 9: Scalability on Rubble: More Available memory with more GPU

training process is irregular and sparse. As a result, although Grendel's distribution strategy may resemble those seen in existing work (e.g., FSDP Zhao et al. (2023)), the details are quite different.

| Method | Rubble | | Building | | MatrixCity Block_All | |
|---|---|---|---|---|---|---|
| | Total Time | Time Decomposition | Total Time | Time Decomposition | Total Time | Time Decomposition |
| CityGS Official | 2.88 hrs | train_coarse: 43min14s
data_partition: 6min5s
9 cells train on 4 A100: 2h3min
Merge point cloud: 59s | 4.57 hrs | train_coarse: 44min3s
data_partition: 17min37s
20 cells train on 4 A100: 3h31min
Merge point cloud: 1min21s | 8.25 hrs | train_coarse: 47min31s
data_partition: 1h28min22s
36 cells train on 4 A100: 5h57min
Merge point cloud: 2min27s |
| CityGS 200K-images | 2.18 hrs | train_coarse: 43min14s
data_partition: 6min5s
9 cells train on 4 A100: 1h20min
Merge point cloud: 57s | 2.22 hrs | train_coarse: 44min3s
data_partition: 17min37s
20 cells train on 4 A100: 1h11min
Merge point cloud: 1min24s | 3.60 hrs | train_coarse: 47min31s
data_partition: 1h28min22s
36 cells train on 4 A100: 1h20min
Merge point cloud: 1min9s |
| Grendel 200K-images (ours) | 0.85 hrs | trained on 4 A100: 51min | 0.90 hrs | trained on 4 A100: 54min | 1.22 hrs | trained on 4 A100: 1h13min |

Table 10: Time Decomposition for experiments in Table 2. CityGaussian employs a divide-and-conquer approach with four steps: coarse training, partitioning the scene into cells, training each cell independently, and merging the resulting point clouds. In contrast, 3DGS on multiple GPUs with Grendel can be run in the same way as the original single-GPU training, simply by allocating additional GPUs.

| Scene (Resolution) | 1 GPU | 4 GPU |
|---|---|---|
| Rubble (3436x4591) | 8.8 img/s | 21.7 img/s |
| Matrixcity Block-all (1080x1920) | 29.2 img/s | 76.8 img/s |
| Bicycle (1275x1920) | 42.6 img/s | 80.3 img/s |

Table 11: 4-GPU Render Speedup with Grendel Compared to Single-GPU Rendering

