# OpenReview forum: "On Scaling Up 3D Gaussian Splatting Training"
_ICLR.cc/2025/Conference — ICLR 2025 Oral_

### Official Review · Reviewer_sLkS · 2024-10-16

**Soundness:** 4
**Presentation:** 2
**Contribution:** 4
**Rating:** 8
**Confidence:** 4

**Summary:**

This paper introduces GrendelGS, which eliminates the bottleneck of 3DGS in parallel training and scaling up. GrendelGS mixes Gaussian-wise and pixel-wise parallelism, so as to fit computation requirements of different stages. It also makes an effort to minimize communication among GPUs and workload imbalance. And it reveals hyperparameter scaling law for 3DGS parallel training.

**Strengths:**

1. Originality.  A reasonable parallel training strategy with novel modification for 3DGS based on valuable insights.
2. Quality. Sufficient experimental validation on effectiveness of the method.
3. Significance. Pioneering and well-optimized parallel training implementation for 3DGS, resolving critical challenges in synchronized parallel 3DGS training. On the one hand, GrendelGS avoids tedious model pertaining or data and primitive partitioning in prior works like VastGaussian or CityGaussian. On the other hand, this work enables efficient training when multiple GPUs are available, especially under instant requirements or large-scale scenes.

**Weaknesses:**

1. While the superior efficiency over the original 3DGS is clearly and sufficiently illustrated through experiments, it would be beneficial to show if this synchronized parallel training mechanism shows higher efficiency over existing open-sourced asynchronized paradigms, such as HierarchyGS, OctreeGS, or CityGS. Note that for a fair comparison, the batch size and overall iterations should be aligned. Considering the limited time for rebuttal, a comparison with one existing method on 2-3 representative large-scale scenes is enough.
2. The Independent Gradients Hypothesis is introduced to support the hyperparameter scaling law. It's reasonable that the hypothesis is true on Rubble or even other large-scale scenes, since the views are extremely sparse and approximately independent. However, for condensed views over small scenes or single objects, the hypothesis may not well suit the reality, especially when the batch size grows.  It seems more reasonable to limit the hypothesis to cases with significant view sparsity, or check if the scaling law brings worse results on small scenes.

**Questions:**

1. The legends in Figure 1 are put in inversed order. A modification seems to be required.
2. In Algorithm 1, the meaning of ET_j is confusing and I didn't find a clear definition. An annotation in the graph can optimize the reading experience.
3. In the "Independent Gradients Hypothesis" section, why can we assume E(g_k)=0 over all views? This hypothesis is not that obvious.

---

> ### Author Response · Authors · 2024-11-20
> **Author Response to Reviewer sLkS (part 1/3)**
>
> Thanks for your detailed review.
>
> > While the superior efficiency over the original 3DGS is clearly and sufficiently illustrated through experiments, it would be beneficial to show if this synchronized parallel training mechanism shows higher efficiency over existing open-sourced asynchronized paradigms, such as HierarchyGS, OctreeGS, or CityGS. Note that for a fair comparison, the batch size and overall iterations should be aligned. Considering the limited time for rebuttal, a comparison with one existing method on 2-3 representative large-scale scenes is enough.
>
> We agree. In the following, we will compare the asynchronous(i.e. divide-and-conquer) training paradigm of CityGS with the 3DGS algorithm running synchronously on our Grendel system. Of the two open-source divide-and-conquer methods, we selected CityGS over HierarchicalGS due to its higher reported PSNR in papers. We perform our comparison on scenes and experimental settings(<https://github.com/DekuLiuTesla/CityGaussian/tree/main/scripts>) from CityGS paper.  We perform all experiments on 4 A100.
> We tested on the following scenes: rubble (downsample 4x), building (downsample 4x), and matrixcity block-all  (downsample to a width of 1600 pixels)
>
> Following the reviewer’s suggestion, to ensure a fair comparison, we fixed the total number of trained images count at 200,000, which is empirically very sufficient for our Grendel system to converge on all three scenes. The official CityGS scripts require an initial coarse training with 30,000 images count, followed by 30,000 images per cell after divide-and-conquer, which will exceed our fixed setting of 200,000 images. We maintained the coarse training at 30,000 images but adjusted the number of images per cell to ensure a total of 200,000 images training for CityGS.
>
> **Rubble Experiments**
> |                                        | PSNR  | SSIM | LPIPS | Total Time | Time Decomposition                                                                                                 |
> |----------------------------------------|-------|------|-------|------------|--------------------------------------------------------------------------------------------------------------------|
> | CityGS                                 | 25.40 | 0.80 | 0.25  | 2h11min    | train_coarse: *43min 14s*. data_partition: *6min5s*. 9 cells train on 4 A100: *1h20min* Merge point cloud: *57s* |
> | Original 3DGS(train on Grendel system) | 27.39 | 0.86 | 0.19  | 51min      | trained on 4 A100: *51min*                                                                                             |
>
> **Building Experiments**
> |                                        | PSNR  | SSIM  | LPIPS | Total Time | Time Decomposition                                                                                                          |
> |----------------------------------------|-------|-------|-------|------------|-----------------------------------------------------------------------------------------------------------------------------|
> | CityGS                                 | 20.32 | 0.73  | 0.30  | 2h13min    | train_coarse: *44min3s*. data_partition: *17min37s*. 20 cells train on 4 A100: *1h11min*. Merge point cloud: *1min24s*. |
> | Original 3DGS(train on Grendel system) | 22.69 | 0.778 | 0.242 | 54min      | trained on 4 A100: *54min*                                                                                                      |

---

> ### Author Response · Authors · 2024-11-20
> **Author Response to Reviewer sLkS (part 2/3)**
>
> **Matrixcity block-all Experiments**
> |                                        | PSNR  | SSIM | LPIPS | Total Time | Time Decomposition                                                                                                         |
> |----------------------------------------|-------|------|-------|------------|----------------------------------------------------------------------------------------------------------------------------|
> | CityGS                                 | 23.68 | 0.70 | 0.42  | 3h36min    | train_coarse: *47min 31s*. data_partition: *1h28min22s*. 36 cells train on 4 A100: *1h20min*. Merge point cloud: *1min9s* |
> | Original 3DGS(train on Grendel system) | 27.33 | 0.86 | 0.21  | 1h13min    | trained on 4 A100: *1h13min*                                                                                                   |
>
> Experimental results indicate that, within the scale of these three scenes, synchronized training of 3DGS using systems like **Grendel can achieve higher training efficiency (both convergence speed and Training Time)** compared to a divide-and-conquer approach.
>
> - The divide-and-conquer approach always involves overlapped partitions(<https://github.com/DekuLiuTesla/CityGaussian/blob/11ece7d88627e3a85f0979b502dd500c526f4367/data_partition.py#L53>), which increases overall computation and memory usage. Additionally, excessive partitioning creates large boundary regions that can introduce artifacts near these boundaries and make convergence more challenging.
> - In contrast, training the original 3DGS algorithm across multiple GPUs with Grendel system-level parallelization circumvents these boundary issues; and this offers a more straightforward approach, as it avoids additional design complexities. Last but not least, Grendel’s system-level parallelization actually complements the divide-and-conquer parallelization, as both coarse training and per-cell training can be further accelerated using the Grendel system, which is especially beneficial for scaling up to even larger scenes.
>
> > The Independent Gradients Hypothesis is introduced to support the hyperparameter scaling law. It's reasonable that the hypothesis is true on Rubble or even other large-scale scenes, since the views are extremely sparse and approximately independent. However, for condensed views over small scenes or single objects, the hypothesis may not well suit the reality, especially when the batch size grows. It seems more reasonable to limit the hypothesis to cases with significant view sparsity, or check if the scaling law brings worse results on small scenes.
>
> Great point! To see if scaling law works well on small scenes, we test on the “garden” scene from Mip-NeRF 360, which has just 185 images (compared to Rubble’s 1657) with camera poses that surround a central object. Intuitively, the gradients should be less sparse compared to Rubble, which is supported by the sparsity plot (<https://anonymous.4open.science/r/grendel-ablations-BD93/ablation_plots/garden_param_featuresdc_sparsity.pdf>). However, we find our proposed hyperparameter scaling law makes batched training highly effective even on Garden. When trained for the same number of epochs, Grendel’s hyperparameter scaling keeps PSNR close to that achieved by batch-size-1 training, while PSNR falls off quickly with bigger batches using default hyperparameters (see <https://anonymous.4open.science/r/grendel-ablations-BD93/ablation_plots/garden_PSNR_vs_scaled_hypers.png>). Grendel works well on most real-world scene sizes.
>
> > The legends in Figure 1 are put in inversed order. A modification seems to be required.
>
> Thank you for the suggestion! We'll update the legend in Figure 1 in the revised version.
>
> > In Algorithm 1, the meaning of ET_j is confusing and I didn't find a clear definition. An annotation in the graph can optimize the reading experience.
>
> Thanks for the suggestion! ET_j represents the Estimated Running Time for computing the j-th pixel block. They are recorded from previous epochs and used to schedule load balancing. We'll revise it in the updated version.

---

> ### Author Response · Authors · 2024-11-20
> **Author Response to Reviewer sLkS (part 3/3)**
>
> > In the "Independent Gradients Hypothesis" section, why can we assume E(g_k)=0 over all views? This hypothesis is not that obvious.
>
> Assuming E[g_k]=0 over all views corresponds to the case when we’re near a local minimum, and the gradients start oscillating near it. This assumption also has the benefit of allowing the (uncentered) second moment of gradients, used by Adam, to agree with the variance. While it is unlikely that E[g_k]=0 holds exactly in practice, we can still empirically verify if the prediction in L312-314 derived from the Hypothesis holds. It says that for a batch of views of size b (with gradients averaged over the views), the second moment, which is the quantity inside the square root in the denominator, should scale following 1/b, times a constant not dependent on b. To this end, we plotted a new analysis of the second moment of gradients over different batch sizes on Rubble (<https://anonymous.4open.science/r/grendel-ablations-BD93/ablation_plots/rubble_second_moment_vs_lr_scaling.pdf>). As batch size increases, the second moment roughly scales by 1/b, since multiplying it by b recovers a relatively flat trend. This empirically shows that in practice, the second moment follows the prediction in L312-314, and that scaling the LR by sqrt(b) approximates the batch-size 1 updates (L316).

---

> ### Comment · Reviewer_sLkS · 2024-11-22
> **Reply to Authors' comments**
>
> I do admire the efforts that the authors made for additional experimental comparison and visualization. Major concerns of mine are solved. I've also read the responses from other reviewers, the additional experiments are rather important for the paper's completeness and should be included in the main body. Another sincere piece of advice is that careless claims and teaser mistakes should be avoided in submission, otherwise, it will leave readers with a bad impression of the rigor and reliability of the research.
>
> Overall, the paper's contribution is undeniably significant to the community. The experimental supports now have been solid and complete. Therefore I have raised my score.

---

> ### Author Response · Authors · 2024-11-22
> **Author Response to Reviewer sLkS**
>
> We're glad our rebuttal has addressed your questions. Following your modification requirements, we have updated the revised version by incorporating the additional experiments into the main body. Due to page limitations, we have moved the ablation study and Figure 13 to the Appendix. In future work, we will include more comparisons with other large-scale reconstruction works. We will also ensure a more thorough review of details moving forward. Thank you again for your valuable feedback!

---

### Official Review · Reviewer_e5kX · 2024-10-28

**Soundness:** 3
**Presentation:** 3
**Contribution:** 3
**Rating:** 8
**Confidence:** 4

**Summary:**

This paper proposes Grendel, a parallelized computation system for 3D Gaussian Splatting. Grendel introduces two parallelization strategies designed for different stages of the 3DGS rendering pipeline: partition by Gaussians and partition by pixels. The partition by Gaussians method distributes Gaussians within the point cloud across multiple GPUs, while partition by pixels assigns pixels across GPUs. Communication between these parallelization schemes is conducted through NCCL all2all. To enhance batch optimization, a learning rate scaling rule is proposed. Experiments demonstrate performance improvements through the utilization of more Gaussians compared to 3DGS approaches in Large scene datasets.

**Strengths:**

1. Proposal of parallelization with respect to different stages of the rendering pipeline of 3DGS is interesting.

2. With the proposed method, training 3DGS on high-resolution images is possible.

3. Batch processing can significantly increase the training speed.

**Weaknesses:**

1. This paper lacks a conclusion and limitations, making it difficult to understand its overall contribution contribution and appear incomplete.

2. Similar to the proposed learning rate scaling, RAdam [1] also suggests a method for variance rectification. It also rectifies the learning rate using variance to enable adaptive learning and, I believe, takes batch size into account. However, the advantage of the proposed method remains unclear.

3. The rendering speed is expected to be slow due to the time required for Gaussian transfers between GPUs, however, the paper does not provide details regarding rendering speed.

4. Based on Fig. 14, it appears that increasing batch size does not decrease the maximum number of Gaussians significantly.

5. As shown in Fig. 11, for datasets that are not high-resolution like Rubble, the performance gain becomes marginal even if the number of Gaussians exceeds what can be handled by a single GPU.

[1] Liu et al., *On the Variance of the Adaptive Learning Rate and Beyond*, ICLR 2020

**Questions:**

1. How much does the rendering speed change?

2. By how much does the actual GPU memory requirement per GPU reduce when using batch processing?

3. What is the storage requirement based on the number of Gaussians?

$~$

Minor comments:

The figure reference is inconsistent in line 173.

---

> ### Author Response · Authors · 2024-11-20
> **Author Response to Reviewer e5kX (part 1/2)**
>
> Thank you for your valuable feedback. We have addressed each comment below and appreciate your thorough review.
>
> > Similar to the proposed learning rate scaling, RAdam [1] also suggests a method for variance rectification. It also rectifies the learning rate using variance to enable adaptive learning and, I believe, takes batch size into account. However, the advantage of the proposed method remains unclear.
>
> RAdam (<https://pytorch.org/docs/stable/generated/torch.optim.RAdam.html>) does not explicitly consider batch size in its hyperparameters. To confirm this, we perform a new analysis regarding hyperparameter scaling with the RAdam optimizer similar to Figure 13, and find that RAdam without hyperparameter scaling does not align well with batch-size-1 updates when using larger batches, similarly to Adam. Additionally, our proposed hyperparameter scaling rule is compatible with RAdam and allows it to align significantly better with batch-size-1 updates. A plot of this new experiment can be found at <https://anonymous.4open.science/r/grendel-ablations-BD93/ablation_plots/radam_rubble_lr_scaling_ablation.png> (compare with Figure 13). Hence, Grendel offers new improvements that uniquely enable hyperparameter-tuning-free distributed 3DGS training.
>
> > The rendering speed is expected to be slow due to the time required for Gaussian transfers between GPUs, however, the paper does not provide details regarding rendering speed.
>
> Our paper focuses on 3DGS training, which is much more memory-intensive and computationally demanding than rendering; therefore, we did not discuss rendering speed in the original paper.
>
> Multi-GPU rendering is actually faster than single-GPU rendering, as we demonstrate in the experiments below (see response to Question 1). We’ll update our manuscript with the rendering results.
>
> > Based on Fig. 14, it appears that increasing batch size does not decrease the maximum number of Gaussians significantly.
>
> Yes! Even increasing the batch size still leaves us with plenty of available memory to accommodate a large number of Gaussians, which is actually a positive sign for our system, rather than a weakness.
>
> > As shown in Fig. 11, for datasets that are not high-resolution like Rubble, the performance gain becomes marginal even if the number of Gaussians exceeds what can be handled by a single GPU.
>
> In theory, it's true that as the number of Gaussians increases, the incremental benefits of adding more will diminish. However, as scenes grow larger, a single GPU's memory will eventually be unable to accommodate the required number of Gaussians—the foundation problem addressed by all large-scale reconstruction methods such as CityGS, HierarchicalGS, and VastGS. For example, the Matrixcity BigCity Aerial scene is 10 times larger than Maxtricity Block-All but shares the same scene type and resolution. Achieving the same PSNR would potentially require 10 times more Gaussians than the 20 million used in Block-All. This amount is far beyond what a single GPU can handle.

---

> ### Author Response · Authors · 2024-11-20
> **Author Response to Reviewer e5kX (part 2/2)**
>
> > How much does the rendering speed change?
>
> Although Gaussian communication can potentially impact rendering time, distributing pixel rendering and 3D-to-2D Gaussian transformations across multiple GPUs significantly reduces computation time. This approach allows our overall render speed to surpass that of single-GPU rendering.
>
> We compare Grendel’s rendering speeds of single-GPU and 4-GPU setups for the Rubble, MatrixCity Block-All, and Bicycle scenes, using our SOTA trained Gaussian parameters. We experiment with various resolutions and scene scales, rendering one image at a time for fair comparison. The 4-GPU setup achieves a speedup of 1.88x to 2.63x, as shown in the table below.
>
> |                                | 1 GPU      | 4 GPU      |
> |--------------------------------|------------|------------|
> | 3436x4591 Rubble               | 8.8 img/s  | 21.7 img/s |
> | 1080x1920 Matrixcity Block-all | 29.2 img/s | 76.8 img/s |
> | 1275x1920 Bicycle              | 42.6 img/s | 80.3 img/s |
>
> > By how much does the actual GPU memory requirement per GPU reduce when using batch processing?
>
> There may be some misunderstanding. With batch processing, the overall GPU memory requirement actually increases because larger batch sizes require more activation memory.
>
> > What is the storage requirement based on the number of Gaussians?
>
> Each Gaussian requires 59 parameters in float32 format, including 48 for spherical harmonics coefficients, 3 for XYZ coordinates, 3 for scales, 4 for rotations, and 1 for opacity. For N million Gaussians, this amounts to N*0.236 GB of storage. In our largest experiment, we generated 40.4 million Gaussians, requiring 59 * 4 * 40.4 million bytes, totaling 9.534 GB of storage.
>
> Note: Training memory consumption is significantly higher than just this Gaussian parameters storage. Beyond the parameters, it includes gradients, optimizer states (e.g., momentum terms), activation memory (like the Z-buffer) during forward and backward passes, and notable memory fragmentation. Combined, these factors can lead to out-of-memory (OOM) errors when constrained to a single GPU.

---

> > ### Comment · Reviewer_e5kX · 2024-11-22
> >
> > I appreciate the author's comments and clarifications. I have raised the score because I agree that the distributed learning model and experiments proposed by the author are important to the community and my main concerns have been addressed. However, I still have concern about the scaling rule.
> >
> > I am not convinced by the claim that a high cosine similarity with a batch size of 1 indicates a good training rule. Moreover, even if the same number of iterations is used, the total number of observed images varies depending on the batch size, which makes a fair evaluation difficult. Furthermore, according to the authors' argument, smaller batch sizes should lead to better performance. However, Figure 8 shows that batch size 16 achieves the best results, and batch size 32 shows no significant difference compared to batch size 16, despite notable differences in cosine similarity. This seems contradictory and needs further clarification.

---

> > > ### Author Response · Authors · 2024-11-22
> > > **Author Response to Reviewer e5kX**
> > >
> > > Thank you for your insightful question. We’d like to clarify that in all of our experiments, the total number of observed images is constant when batch size varies. We ensure a fair comparison by scaling down the total number of updates when batch size increases, so the total number of images is the same. Notably, this is distinct from the methodology of some concurrent works such as “MVGS: Multi-view-regulated Gaussian Splatting for Novel View Synthesis”, which increases the number of total observed images by the same factor as the number of views per batch (simply called “batch size” in our work). Please refer to our response to reviewer VE39 for a more detailed discussion with quantitative results that support the unique contribution of our scaling rule. We will revise our text to more explicitly convey the important point that our ablations on the batch size are **fair comparisons** in that the number of observed images is held equal.
> > >
> > > We use the cosine similarity and the norm ratio as an analytical tool to examine optimization under different batch sizes. While high similarity and a norm ratio close to one are sufficient for generalizing batch-size-1 optimization to bigger batch sizes, it may not be necessary. Indeed, as you pointed out, batch size 32 in Figure 8 achieves similar, albeit slightly reduced PSNR than does batch size 16, despite the differences in cosine similarity (Figure 13 (a)). This suggests that even though batch size 32 diverges more quickly from batch-size-1 updates, it nonetheless finds decent solutions that have comparable PSNRs. The advantage offered by our scaling rule over the baseline with default Adam hyperparameters is more elucidated more clearly in this plot (<https://anonymous.4open.science/r/grendel-ablations-BD93/ablation_plots/garden_PSNR_vs_scaled_hypers.png>), where we show Grendel’s scaling rule keeps PSNR roughly invariant to the batch size. Please refer to our discussion with reviewer sLkS for further details. Jointly showing analytical and empirical (PSNR) experiments further highlights the effectiveness of Grendel’s scaling rule, thus we plan on showing more PSNR-based ablations in the paper.
> > >
> > > We are more than willing to continue engaging in in-depth discussions. If you have any further questions, please feel free to comment at any time.

---

> > > > ### Comment · Reviewer_e5kX · 2024-11-23
> > > >
> > > > The results of https://anonymous.4open.science/r/grendel-ablations-BD93/ablation_plots/garden_PSNR_vs_scaled_hypers.png, show that fixing epochs does not guarantee convergence (training iterations are reduced), and scaling the gradient helps, but when parameters are trained with the same number of times (M $\times$ Epochs experiments), it seems that a large batch size optimizes in a more accurate direction. For this reason, comparing the similarity with a batch size of 1 seems inappropriate (the similarity seems to be different due to the different training speed when comparing according to the training images). And is it possible to provide an experiment on RAdam similar to https://anonymous.4open.science/r/grendel-ablations-BD93/ablation_plots/PSNR_vs_num_views_vs_num_epochs.png? I wonder if other types of optimizers are also guaranteed similar convergence by the scaling rule.

---

> > > > > ### Author Response · Authors · 2024-11-24
> > > > >
> > > > > The interpretation that a large bath size optimizes in a more accurate direction when parameters are updated the same number of times (M × Epochs) is accurate. However, the training also runs for roughly M times as long as the number of observed training images goes up by M times. This differs from our setting where the goal is to enable hyperparameter-tuning-free training for different batch sizes whiling keeping the total number of observed images (epochs) to limit training times. To this end, it is approriate to compare training at the same number of observed training images, as we do in Figure 12 so that the different training speeds can be revealed. Indeed, <https://anonymous.4open.science/r/grendel-ablations-BD93/ablation_plots/PSNR_vs_num_views_vs_num_epochs.png> shows that large-batch-size training obtains poorer PSNRs with equal epochs because they train slower. While one could train for longer and get improved PSNRs, it would defeat the purpose of the speedup offered by distributed training. We do not consider this to be a practical setting.
> > > > > We are happy to provide an experiment on RAdam similar to <https://anonymous.4open.science/r/grendel-ablations-BD93/ablation_plots/PSNR_vs_num_views_vs_num_epochs.png>. We make a new plot at <https://anonymous.4open.science/r/grendel-ablations-BD93/ablation_plots/radam_PSNR_vs_scaled_hypers.png>. When we swap Adam for RAdam while keeping all else the same, higher number of views (batch size) achieves reduced PSNRs than does Adam in the constant-epochs setting. (notice the re-scaled y-axis). Our hyperparameter scaling rule remains highly effective and recovers most of the reduction in PSNR.
> > > > > We expect that the scaling rule to be effective on all Adam-type optimizers that scale gradient updates by the inverse-square-root of second moments.

---

> > > > > > ### Comment · Reviewer_e5kX · 2024-11-25
> > > > > >
> > > > > > I appreciate the provision of additional experiments. This resolved an unclear point I had in the experiment. I agree with the author's rebuttal and have raised the score accordingly.

---

> > > > > > > ### Author Response · Authors · 2024-11-25
> > > > > > >
> > > > > > > We're glad the additional experiments addressed your questions. Thank you again for reviewing our work and providing thoughtful feedback.

---

### Official Review · Reviewer_smer · 2024-11-01

**Soundness:** 3
**Presentation:** 4
**Contribution:** 4
**Rating:** 8
**Confidence:** 5

**Summary:**

This paper proposes a parallel training pipeline for 3D Gaussian Splatting to be trained on up to 32 GPUs, instead of 1 GPU only. This significantly improves the limit of 3DGS in terms of primitive count, scene scale, scene detail, and training speed. This paper splits the 3DGS training process into per-Gaussian stage and per-pixel stage. In per-Gaussian stage, each GPU has its own Gaussians to calculate the Gaussian transformations, mostly 3D to 2D transformation and color evaluation. In per-pixel stage, each GPU is in charge of a set of pixels for rendering, and performs a sparse communication to fetch the required Gaussians from other GPUs. Lastly, a backward propagation is carried out in a similar manner to pass back the gradients. As a result, the proposed method demonstrates a significantly higher number of Gaussians supported, much faster training speed, and a moderate increase in rendering quality.

**Strengths:**

**Motivation**
* This paper has a strong motivation of expanding the training of 3DGS into multiple GPUs/machines. The vanilla 3DGS has a limited number of primitives supported due to the GPU memory limit, but the exceptional performance of 3DGS indicates its great usefulness in large-scale reconstruction. This paper is monumental in providing support in terms of parallel training.

**Method**
* This paper correctly identifies the per-Gaussian and per-pixel stages of the training pipeline of 3DGS. Since it split the set of Gaussians onto each GPU, the first per-Gaussian stage is relatively simple. For the per-pixel stage, the paper provides intuitive and empirical support for the locality property of 3DGS for each pixel. The paper cleverly leverages this property to design a sparse communication to transport the 3DGS across GPU at a low speed and bandwidth sacrifice. This enables the distributed training of 3DGS.
* This paper rigorously analyzed the scaling property of the 3DGS training process, and carefully designed a hyper-parameter setting paradigm based on the independent gradient assumption and empirical support.
* To further improve the utilization of all GPUs, the load balancing algorithm is introduced to ensure each GPU can adaptively adjust their workload during training, because the number of Gaussians required for each region of the image cannot be predetermined.

**Experiments**
* This paper conducts extensive experiments across various scales of scenes under different experiment settings. The experiment result are well presented to demonstrate the improvement in primitive count, training speed, and rendering quality.

**Weaknesses:**

**Method**
* The paper follows the vanilla 3DGS design and does not consider the Level of Detail support. Since this paper focuses on the large-scale reconstruction, LOD is essential for downstream applications. It is not clear how the proposed method can support LOD.

**Experiment**
* The quality improvement is not as significant as the number of primitives or the training speed. However, I would like to argue that this is because of the lack of an appropriate dataset to evaluate a very large-scale reconstruction, which limit the performance of this paper.
* Although the training speed improvement is very significant with this parallel training, large scenes require very time-consuming pose estimation to be done first. The overall training time including the pose estimation might not be improving as much as the reported improvement. This is not the problem of this paper, but does weaken the impact of this paper.

**Questions:**

1. Can the proposed method be applied to 3DGS with LOD design?
2. How does the training across different machines instead of different GPUs affect the performance of the proposed paper?

---

> ### Author Response · Authors · 2024-11-20
> **Author Response to Reviewer smer**
>
> Thanks for your careful review.
>
> > Although the training speed improvement is very significant with this parallel training, large scenes require very time-consuming pose estimation to be done first. The overall training time including the pose estimation might not be improving as much as the reported improvement. This is not the problem of this paper, but does weaken the impact of this paper.
>
> That is a great point! We anticipate that with advancements in hardware like LiDAR, we will be able to obtain camera poses more easily. Combined with techniques like pose optimization, these could allow us to skip the slow COLMAP pose estimation. But these are indeed beyond the scope of this paper. In our future work, we will try accelerating pose estimation.
>
> > Can the proposed method be applied to 3DGS with LOD design?
>
> That’s a great question! Yes, our distribution strategy can, in principle, be effectively integrated with Level of Detail (LoD) techniques.
>
> Our system design is based on the fact that 3D Gaussian Splatting (3DGS) involves both Gaussian-wise and Pixel-wise computations, allowing us to partition either primitives (Gaussians) or images across GPUs. Even with LoD methods, like those in OctreeGS or HierarchicalGS, the core computations remain primarily primitive-wise and pixel-wise, enabling similar parallelization. In OctreeGS, primitives serve as anchors at different levels, while in HierarchicalGS, they correspond to Gaussians associated with intermediate and leaf nodes.
>
> However, implementing LoD brings additional challenges that require further exploration. For example, LoD depends on a hierarchical (tree) structure, where rendering requires determining a specific cut within this hierarchy, adding complexity to creating an efficient distribution strategy for these specialized data structures and computational steps. Supporting LoD is a promising direction for future work.
>
> > How does the training across different machines instead of different GPUs affect the performance of the proposed paper?
>
> You bring up a key point in system scalability design. Cross-machine distribution affects system throughput due to slower network speeds compared to NVLink within a single machine, which limits cross-machine scalability. As shown in Figures 7 and 8 of our paper, we scale up to 32 GPUs across 8 machines: while the 4K rubble scene scales efficiently (32 GPUs achieve a 5.2x speedup over 4 GPUs), the smaller Train scene struggles to utilize additional resources effectively (16 GPUs achieve only a 1.64x speedup over 4 GPUs). In our future work, we'll focus on refining our distribution strategy to improve cross-node scalability. Additionally, from the memory perspective, multi-machine setups can provide even more GPU memory, enabling the generation of a greater number of Gaussians.

---

> > ### Comment · Reviewer_smer · 2024-11-22
> >
> > I appreciate the extra experiments carried out by the author and the detailed explanation for my questions. I still believe the paper carries sufficient technical innovation and good application value, but I also agree with the other reviewers that the evaluation and comparison against other baselines need to be done carefully.
> >
> > I look forward to the extension of the method onto LoD solutions, and I would like to keep my rating.

---

> ### Author Response · Authors · 2024-11-22
> **Thank you once again for your positive feedback on our paper!**
>
> Thank you once again for your positive feedback on our paper! We recognize the importance of thorough baseline evaluations and will prioritize this moving forward.
>
> Supporting LoD is indeed a critical direction, and we are committed to addressing it soon.

---

### Official Review · Reviewer_VE39 · 2024-11-03

**Soundness:** 3
**Presentation:** 4
**Contribution:** 4
**Rating:** 8
**Confidence:** 5

**Summary:**

This paper, named Grendel, presents a system to enable the distributed training of 3DGS. By leveraging the spatial locality and discovering the Gaussian intersection on image pixels, Grendel enables transferring Gaussians with sparse all-to-all communication while balancing the communication overloads. The load balancing can be simply adjusted according to the time estimation of rendering a pixel in previous iterations. While increasing the batch size needs adjusting the hyperparameters for optimizers, Grendel proposed an automatic hyperparameter scaling rule to enable the hyperparameter-tuning-free batched 3DGS training. Experiments conducted on the small-scale Mip-NeRF360, Tansk-and-Temples, and DeepBlending dataset, the large-scale Mega-NeRF dataset, and the large-scale MatrixCity data validated the effectiveness of the proposed method.

**Strengths:**

(1) This work enables the distributed training of 3DGS, which can be very useful to accelerate the training of 3DGS.

(2) The paper is well-written and easy to understand.

(2) This work did exhaustive experiments on very large-scale datasets (e.g., the MatrixCity dataset)

**Weaknesses:**

(1) The related works are put into the last section of the paper, which is weird to me. It lacks discussion of some existing distributed methods, such as **DOGS** (Chen and Lee, NeurIPS 2024), and RetinaGS(Li, et, al, arXiv 2024).

(2) In Sec.6 Related Works, The discussion of VastGaussian, CityGaussian, and Hierarchical Gaussian is wrong. They do not merge the resulting images. Instead, they merge the sub-models. And the authors also claimed that "None of these systems can consider a full large-scale scene directly and achieve the same quality as ours". This is ***over-claimed and does not respect existing works since they did not compare to any of these methods*** but only tried different configurations of their Grendel methods.

(3) Baselines are missing in the current version. It is a good system work to me. However, over-claiming something in the paper won't make it a better work. I would like to see a fair comparison of the method to more baselines (original 3DGS, VastGaussian, CityGaussian, DOGS, etc.) and raise my score if they do so.

**Questions:**

(1) At line 206, the method splits an image into 16x16 pixel blocks. I'm curious whether the patch size of image blocks can influence the performance of image rendering quality.

(2) Since this method enables training 3DGS from multiple views, it can gather gradients from multiple views and then better constrain the training of 3DGS (as has been shown in the paper *MVGS: Multi-view-regulated Gaussian Splatting for Novel View Synthesis*). However, I saw no performance gains using the distributed training. In lines 312-323, the paper claimed that "even though some cameras share similar poses, a set of random cameras generally observe different parts of a scene, hence the gradients in a batch are mostly sparse and can be thought of as roughly independent.". I think that might be the reason that the distributed training does not improve the rendering quality. Can the authors try to explain/discuss this further? For example, are there any non-random batch-splitting methods that can be applied to improve the performance?

---

> ### Author Response · Authors · 2024-11-20
> **Author Response to Reviewer VE39 (Part 1/4)**
>
> Thanks for your insightful comments, we respond to the comments below:
>
> > (1) The related works are put into the last section of the paper, which is weird to me. It lacks discussion of some existing distributed methods, such as DOGS (Chen and Lee, NeurIPS 2024), and RetinaGS(Li, et, al, arXiv 2024).
>
> Thank you for suggesting to include DoGS and RetinaGS in the related work. Since both works are closed-source, we can't perform a quantitative comparison at the moment and will just discuss the differences between these works and ours below.
>
> - *Comparison to DoGS:* Grendel, using system-level parallelization, does not modify the original 3DGS algorithm, thus preserving the same convergence characteristics as the original algorithm run on a single GPU. In contrast, DoGS changes the training algorithm to use ADMM distributed optimization, averaging Gaussians shared across partitioning boundaries once every 100 iterations. For these shared Gaussians, DoGS essentially performs asynchronous training which may impact the convergence rate.
>
> - *Comparison to RetinaGS:* RetinaGS also does not modify the original 3DGS algorithm but uses a different parallelization strategy than Grendel. In RetinaGS, every GPU renders the same whole image using a local partition of Gaussians and then merges the rendered images together.  Doing so causes redundant rendering computation as a pixel may be beyond the opacity saturation depth on many GPUs but is still rendered by them.  Additionally, RetinaGS’s parallelization strategy can have significant load imbalance in rendering, esp. for small batch sizes.
>
>
> > (2) In Sec.6 Related Works, The discussion of VastGaussian, CityGaussian, and Hierarchical Gaussian is wrong. They do not merge the resulting images. Instead, they merge the sub-models.
>
> Thank you for pointing out an error in our submitted version. We will fix this in the revised version.
>
> > And the authors also claimed that "None of these systems can consider a full large-scale scene directly and achieve the same quality as ours". This is over-claimed and does not respect existing works since they did not compare to any of these methods but only tried different configurations of their Grendel methods.
>
> **We apologize for this overclaim and will remove it in the revision.** By saying “none of these systems consider a full-scale scene directly”, we meant to emphasize that Grendel does not modify the original 3DGS algorithm, compared to the divide-n-conquer works that change the 3DGS optimization pipeline. In the revision, we will give a more careful discussion of these related work. Specifically, we’ll point out the following:
> 1. **Keeping the original 3DGS algorithm intact while relying on system-level parallelization avoids the boundary issue compared to divide-n-conquer approach which has to address the boundary issue.** Our experiments, described below, show that Grendel can achieve quality that is better or comparable to CityGS.
> 2. **Grendel can work together with divide-n-conquer algorithms.** For example, the initial coarse training step in both CityGS and HierarchicalGS can leverage Grendel for multi-GPU acceleration.   On the other hand, Grendel can incorporate certain new algorithmic innovations proposed in divide-n-conquer works, such as modifying densification for sparser views (HierarchicalGS), applying appearance embeddings to handle exposure changes (VastGS), or employing level-of-detail techniques to enhance rendering efficiency (CityGS, HierarchicalGS).
> 3. **Grendel can benefit scenarios other than large-scale scene reconstruction.** For example, Figures 8 and 9 demonstrate our speedup on small-scale scenes, including Tanks & Temple, DeepBlending, and Mip360—scenarios beyond the scope of the three large-scale reconstruction papers. Additionally, our pixel-wise parallelization also enables us to accelerate high-resolution image reconstruction, with the corresponding speedup shown in Figure 7.

---

> ### Author Response · Authors · 2024-11-20
> **Author Response to Reviewer VE39 (Part 2/4)**
>
> > (3) Baselines are missing in the current version. It is a good system work to me. However, over-claiming something in the paper won't make it a better work. I would like to see a fair comparison of the method to more baselines (original 3DGS, VastGaussian, CityGaussian, DOGS, etc.) and raise my score if they do so.
>
> For comparison with the original 3DGS training on a single GPU, see Figures 9, 8 and 11. We want to emphasize again that our system-level parallelization remains entirely faithful to the original 3DGS algorithm. With the same hyperparameter settings, the training loss and gradients match at every step between the original single-GPU training and Grendel’s multi-GPU training. Therefore, we can achieve speedups without affecting test PSNR, as illustrated in Figures 9 and 8. In Figure 11, we also achieve higher test PSNR by generating additional Gaussians beyond single-GPU memory, further unlocking the potential of the original 3DGS.
>
> We provide the experiments comparison with CityGS below. We could not compare it with DoGS,  RetinaGS, VastGS because they are not open sourced. Due to the time limit, we have not compared against HierarchicalGS, but we note that CityGS has better reported quality than HierarchicalGS.
>
> Our experiments run on a machine with 4 A100 GPUs using these scenes: Rubble (1657 images, downsample 4x), Building (1940 images, downsample 4x), and matrixcity block-all (5620 images, downsampled to a width of 1600 pixels). Below are our comparison points.
> 1. *CityGS Official.* This is CityGS trained using the authors’ official script (<https://github.com/DekuLiuTesla/CityGaussian/tree/main/scripts>) and their configuration (<https://github.com/DekuLiuTesla/CityGaussian/tree/main/config>). The PSNR numbers shown below are comparable to those reported in the CityGS paper. Rubble, Building, and MatrixCity Block-All are trained on 300000, 630000, and 1110000 images, respectively. Note that the number of images in the datasets are far fewer than the number of images trained.
> 2. *Original 3DGS(trained using Grendel).* We run the original 3DGS algorithm using Grendel. We use the same hyperparameters (e.g. positional learning rate) as used by CityGS-Official when applicable. We empirically observe that the original 3DGS can converge after training 200,000 images for the dataset used, so we stop the training after 200,000 images.
> 3. *CityGS 200K-images.* For a fair comparison with Grendel, we also train CityGS and stop after 200,000 images, referred to as `CityGS 200k-images`.
>
> **Rubble Experiments**
> |                                        | # of Trained Images | PSNR  | SSIM | LPIPS | Total Time | Time Decomposition                                                                                                 |
> |----------------------------------------|---------------------|-------|------|-------|------------|--------------------------------------------------------------------------------------------------------------------|
> | CityGS Official                        | 300,000             | 25.88 | 0.81 | 0.23  | 2h53min    | train_coarse: *43min 14s*. data_partition: *6min5s*. 9 cells train on 4 A100: *2h3min*. Merge point cloud: *59s*  |
> | CityGS 200k-images                     | 200,000             | 25.40 | 0.80 | 0.25  | 2h11min    | train_coarse: *43min 14s*. data_partition: *6min5s*. 9 cells train on 4 A100: *1h20min* Merge point cloud: *57s* |
> | Original 3DGS(train on Grendel system) | 200,000             | 27.39 | 0.86 | 0.19  | 51min      | trained on 4 A100: *51min*                                                                                             |
>
> **Building Experiments**
> |                                        | # of Trained Images | PSNR  | SSIM  | LPIPS | Total Time | Time Decomposition                                                                                                          |
> |----------------------------------------|---------------------|-------|-------|-------|------------|-----------------------------------------------------------------------------------------------------------------------------|
> | CityGS Official                        | 630,000             | 22.14 | 0.784 | 0.241 | 4h34min    | train_coarse: *44min3s*. data_partition: *17min37s*. 20 cells train on 4 A100: *3h31min*. Merge point cloud: *1min21s*.     |
> | CityGS 200k-images                     | 200,000             | 20.32 | 0.73  | 0.30  | 2h13min    | train_coarse: *44min3s*. data_partition: *17min37s*. 20 cells train on 4 A100: *1h11min*. Merge point cloud: *1min24s* |
> | Original 3DGS(train on Grendel system) | 200,000             | 22.69 | 0.778 | 0.242 | 54min      | trained on 4 A100: *54min*                                                                                                      |

---

> ### Author Response · Authors · 2024-11-20
> **Author Response to Reviewer VE39 (Part 3/4)**
>
> **Matrixcity block-all Experiments**
>
> |                                        | # of Trained Images | PSNR  | SSIM  | LPIPS | Total Time | Time Decomposition                                                                                                          |
> |----------------------------------------|---------------------|-------|-------|-------|------------|-----------------------------------------------------------------------------------------------------------------------------|
> | CityGS Official                        | 1,110,000           | 27.41 | 0.864 | 0.205 | 8h15min    | train_coarse: *47min 31s*. data_partition: *1h28min22s*. 36 cells train on 4 A100: *5h57min*. Merge point cloud: *2min27s* |
> | CityGS 200k-images                     | 200,000             | 23.68 | 0.70  | 0.42  | 3h36min    | train_coarse: *47min 31s*. data_partition: *1h28min22s*. 36 cells train on 4 A100: *1h20min*. Merge point cloud: *1min9s*  |
> | Original 3DGS(train on Grendel system) | 200,000             | 27.33 | 0.859 | 0.205 | 1h13min    | trained on 4 A100: *1h13min*                                                                                                    |
>
> Observations
> - **Grendel achieves test PSNR comparable to, or surpassing those of CityGS-Official**. **Grendel is much faster**—achieving 3x-7x speed improvements over CityGS-Official. Our experience also shows that **Grendel is simpler to use**. CityGS requires running several separate procedures each of which requires hyperparameter tuning.  By contrast, we can run Grendel in the same way as the original 3DGS by simply allocating more GPU via Slurm.
>
> - We note that the settings used in our paper are more challenging than those used in the above experiments  which are the same as in the CityGS paper.  In particular, our paper uses the original resolution (e.g., rubble 4591x3436), while CityGS (and HierarchicalGS, DoGS) downsamples (e.g., rubble 1152x864). We encountered out-of-memory issues when running CityGS on Rubble using the original resolution.

---

> ### Author Response · Authors · 2024-11-20
> **Author Response to Reviewer VE39 (Part 4/4)**
>
> > (1) At line 206, the method splits an image into 16x16 pixel blocks. I'm curious whether the patch size of image blocks can influence the performance of image rendering quality.
>
> No, changing the pixel block size doesn’t alter any pixel values in the final rendered image. The block size only impacts load distribution across GPUs, as smaller blocks allow for finer control over how work is distributed. All systems designs in Section 3 maintain the identical rendering calculation; the only difference lies in which GPU performs each calculation.
>
> > (2) Since this method enables training 3DGS from multiple views, it can gather gradients from multiple views and then better constrain the training of 3DGS (as has been shown in the paper MVGS: Multi-view-regulated Gaussian Splatting for Novel View Synthesis). However, I saw no performance gains using the distributed training. In lines 312-323, the paper claimed that "even though some cameras share similar poses, a set of random cameras generally observe different parts of a scene, hence the gradients in a batch are mostly sparse and can be thought of as roughly independent.". I think that might be the reason that the distributed training does not improve the rendering quality. Can the authors try to explain/discuss this further? For example, are there any non-random batch-splitting methods that can be applied to improve the performance?
>
> Great question! Our method can, and indeed does, gather gradients from multiple views during each iteration in a way exactly equivalent to the paper “MVGS: Multi-view-regulated Gaussian Splatting for Novel View Synthesis.” Equation (3) in the MVGS paper states that each iteration consists of gradients from M random views (see also their code at <https://github.com/xiaobiaodu/MVGS/blob/master/train.py#L89>), which is equivalent to our approach.
> MVGS reports improved PSNR in the multi-view regulated training ablation in MVGS Fig. 5 because of its differing testing methodology. When they increase M – the number of views per iteration – the number of training iterations remains unchanged (<https://github.com/xiaobiaodu/MVGS/blob/master/run_360.py#L26>). Effectively, the number of training epochs multiplies by M. By contrast, when the batch size is increased in Grendel, we scale down the number of iterations to hold the number of epochs constant. To verify this experimentally, we reproduced MVGS’s multi-view regulated training ablation experiment by adding multi-view regulated training to the INRIA’s official 3D GS code. We test with M’s in {1, 5, 10, 20} on the "room", "counter", "kitchen", and "bonsai" scenes from Mip-NeRF 360 and show PSNR numbers in the table below.
>
> | M | PSNR (M × Epochs)   | PSNR (Constant Epochs) | PSNR (Constant Epochs, Scaled Hypers) |
> |---|---|---|---|
> | 1 | 30.83 | 30.92 | 31.04 |
> | 5 | 31.33 | 30.50 | 30.97 |
> | 10 | 31.48 | 29.85 | 30.84 |
> | 20 | 31.59 | 28.68 | 30.53 |
>
> With a constant number of epochs (our methodology, but without the hyperparameter scaling), image quality degrades as M increases. Only when the number of iterations is unchanged in “M × Epochs”, (MVGS methodology), do we observe image quality gains similar to MVGS Fig. 5. A visual plot can be found at <https://anonymous.4open.science/r/grendel-ablations-BD93/ablation_plots/PSNR_vs_num_views_vs_num_epochs.png>.  The MVGS setup (“M × Epochs”) obtains slightly improved PSNRs with increasing M, but each of the runs takes roughly M times as long as the M=1 case.
>
> While this new experiment shows that multi-view regulated training degrades image quality with equal training epochs, it is unsurprising. Importantly, Grendel effectively addresses this with a new hyperparameter scaling rule in Section 4. With this technique, “Constant Epochs, Scaled Hypers” effectively alleviates the image quality degradation. We will include this discussion in the updated manuscript.
>
> Additionally, we believe using non-random batch-splitting methods to improve image quality is an excellent venue for research. While this work focuses on enabling efficient, hyperparameter-tuning-free distributed training of 3DGS, we will consider this as a future extension to Grendel.

---

> ### Comment · Reviewer_VE39 · 2024-11-21
> **Thanks for the reply**
>
> The authors write a very strong rebuttal. The major concerns of mine are solved, especially the authors added experiments to compare against CityGS and show that Grendel surpasses or comparable to CityGS on the rubble scene, building scene, and the MatrixCity dataset. In addition, the authors did experiments to show their distributed training can improve the rendering quality, which is equivalent to the multiple-view training strategy of MVGS. I really appreciate the authors for providing experiments like this.
>
> Overall, I think this work is very helpful to both the academia community and industrial community and I would like to raise my score to accept with some part of the paper being revised properly.

---

> ### Author Response · Authors · 2024-11-21
> **Thanks for your thoughtful feedback and for raising your score!**
>
> Thank you very much for your thoughtful feedback and for raising your score. Your suggestions are invaluable in helping us improve the paper. We sincerely appreciate your recognition of our efforts to address the major concerns. We will thoroughly revise the paper to incorporate all of your feedback.

---

### Author Response · Authors · 2024-11-27
**Revision Summary**

We sincerely thank all reviewers for their active engagement and constructive feedback. Following the discussion with reviewers, we have updated the manuscript, where the changed text is highlighted in blue. Here we summarize the revision as follows:
1. We have added experiments comparing Grendel’s system-level parallelization with CityGaussian’s divide-and-conquer approach in the Evaluation Section of the main body, which is suggested by reviewers VE39 and sLKS. Due to page limitations, the time decomposition results for CityGaussian are included in the Appendix C.
2. The Ablation Study subsection and Figure 13 (Visualization: Gaussian Quantity vs. Reconstruction Quality) in the Evaluation Section have been moved to the Appendix C due to page limitations.
3. The results of render speedup experiments using Grendel’s multi-GPU parallelization have been added to the Appendix C.4, as suggested by reviewer e5kX.
4. We have rewritten the discussion on related works about other large-scale scene reconstruction approaches in the Related Works Section, as suggested by reviewer VE39. The revised version highlights that our system-level parallelization is complementary to divide-and-conquer approaches.
5. We have incorporated DoGS and RetinaGS as related works relevant to distributed training for 3DGS in the Related Works Section, as suggested by reviewer VE39.
6. The discussion on related works for distributed training in neural networks has been moved to the Appendix D to accommodate page restrictions.
7. We have fixed the mislabeling issue for the scene names in Figure 1, as pointed out by reviewer sLKS.
8. Annotation for the term ET_j  has been added to Algorithm 1 to enhance clarity, as suggested by reviewer sLKS.

We are thrilled that all reviewers acknowledge the significance of our multi-GPU distributed 3DGS training system for the community. As we plan to open source Grendel, this work will allow more users to build upon our work and further create more efficient, scalable infrastructure for the 3DGS/AI community. Following the discussion, we polished the text to fix typos and increase the accuracy of our claims. We plan to add new ablations presented in the rebuttal to the camera-ready text to extend the discussion around Grendel’s unique contributions. We greatly appreciate the reviewers’ meticulous efforts and valuable feedback!

---

### Meta-Review · Area_Chair_WF9E · 2024-12-20

**Metareview:**

This paper presents a parallel training method for 3D Gaussian splatting, which significantly improves the training speed and working scene scale. The paper is strongly motivated for scaling the 3DGS. Although the quality improvement is rather limited, the method has a strong merit in rigorous analysis of the scaling property of 3DGS across GPUs, and the effectiveness of the proposed method is significant. The four expert reviewers are consistently happy about the paper, resulting in all positive ratings. The AC agreed with the reviewers' opinions and recommended the acceptance of the paper.

**Additional Comments On Reviewer Discussion:**

The reviewers pointed out the missing literature that were relevant to the submitted paper. These were discussed in the author's response, and the paper was appropriately revised accordingly. In addition, the reviewers suggested additional experiments to further verify the effectiveness of the proposed method. This also is adequately taken care by the authors.
In general, the reviewers were positive from the intial review. The reviewers and authors have communicated thoroughly to fix remaining issues about the paper.

---

### Decision · Program_Chairs · 2025-01-22

Accept (Oral)